# A prevalence-based transmission model for the study of the epidemiology and control of soil-transmitted helminthiasis

**Nyuk Sian Chong**[1], **Robert J. Hardwick**[2], **Stacey R. Smith?**[3]*, **James E. Truscott**[2], **Roy M. Anderson**[2]

**1** Faculty of Ocean Engineering Technology and Informatics, Universiti Malaysia Terengganu, Terengganu, Malaysia, **2** London Centre for Neglected Tropical Disease Research, Department of Infectious Disease Epidemiology, St. Mary's Campus, Imperial College London, London, United Kingdom, **3** Department of Mathematics and Faculty of Medicine, The University of Ottawa, Ottawa, ON, Canada

* stacey.smith@uottawa.ca

**Data Availability Statement:** All relevant data are within the paper.

## Abstract

Much effort has been devoted by the World Health Organization (WHO) to eliminate soil-transmitted helminth (STH) infections by 2030 using mass drug administration targeted at particular risk groups alongside the availability to access water, sanitation and hygiene services. The targets set by the WHO for the control of helminth infections are typically defined in terms of the prevalence of infection, whereas the standard formulation of STH transmission models typically describe dynamic changes in the mean-worm burden. We develop a prevalence-based deterministic model to investigate the transmission dynamics of soil-transmitted helminthiasis in humans, subject to continuous exposure to infection over time. We analytically determine local stability criteria for all equilibria and find bifurcation points. Our model predicts that STH infection will either be eliminated (if the initial prevalence value, $y(0)$, is sufficiently small) or remain endemic (if $y(0)$ is sufficiently large), with the two stable points of endemic infection and parasite eradication separated by a transmission breakpoint. Two special cases of the model are analysed: (1) the distribution of the STH parasites in the host population is highly aggregated following a negative binomial distribution, and (2) no density-dependent effects act on the parasite population. We find that disease extinction is always possible for Case (1), but it is not so for Case (2) if $y(0)$ is sufficiently large. However, by introducing stochastic perturbation into the deterministic model, we discover that chance effects can lead to outcomes not predicted by the deterministic model alone, with outcomes highly dependent on the degree of worm clumping, $k$. Specifically, we show that if the reproduction number and clumping are sufficiently bounded, then stochasticity will cause the parasite to die out. It follows that control of soil-transmitted helminths will be more difficult if the worm distribution tends towards clumping.

**Funding:** NSC: Ministry of Education of Malaysia and the Faculty of Ocean Engineering Technology and Informatics; Universiti Malaysia Terengganu; UMT for the TAPE-RG research grant support (project vot: 55228); UK Medical Research Council and Department for International Development SRS: NSERC Discovery and Alliance Grants RJH, JET, RMA: Bill and Melinda Gates Foundation via the DeWorm3 (OPP1129535) award to the Natural History Museum in London; UK Medical Research Council and Department for International Development The funders had no role in study design, data collection and analysis, decision to publish, or preparation of the manuscript.

**Competing interests:** The authors have declared that no competing interests exist.

# Introduction

Soil-transmitted helminths (STHs) are intestinal worm parasites, which are transmitted to humans through contaminated soil via eggs or larvae present in faecal material deriving from infected individuals who harbour reproductively mature female worms [1]. Transmission is prevalent in areas with poor hygiene and sanitation [2]. The primary species of STH that infect humans are roundworm (*Ascaris lumbricoides*), whipworm (*Trichuris trichiura*) and hookworm (*Necator americanus* and *Ancylostoma duodenale*) [3]. Humans become infected after they come into contact with contaminated soil, objects or surfaces, or by ingesting contaminated food or drink with parasite eggs or larvae [4].

The World Health Organization (WHO) [5] has reported that there are more than 1.5 billion people worldwide who suffer from STH infection [5]. Most of the infected cases are found in sub-Saharan Africa, East Asia, the Americas and China [6]. Although STH infections are not a major cause of mortality, they impair child growth, particularly cognitive and physical development [7]. They may also cause malnutrition and intestinal clinical manifestations and can generate social stigma [8]. STH infections not only cost billions of dollars in interventions annually but they also lead to poor health and a rise in the disability-adjusted life years (DALYs) [9]. They are a significant health burden, especially in people who are living in poverty, even though they are treatable and preventable [8].

Eradicating morbidity due to STH infection in children by 2030 is one of the global targets set by the WHO in the 2021–30 NTDs (neglected tropical diseases) Road Map [10]. The WHO aims to ensure accessibility to basic sanitation and hygiene to prevent infections and reinfections in STH endemic regions by 2030 [11]. Through behavioural interventions such as shoe wearing, hand-washing and waste/excreta management and the WASH (water, sanitation and hygiene) program, global access to clean and safe water, adequate sanitation and hygiene can be achieved, which will promote healthy living, improve socio-economic development and reduce poverty [12]. Soil-transmitted helminthiasis can be treated by either 400mg of albendazole or 500mg of mebendazole. These medicines are typically donated by GlaxoSmithKline and Johnson & Johnson to the WHO for use in endemic regions, can be dispensed by non-medical personnel, are effective and generate very few side effects [13–17]. The WHO proposes periodic treatment for all at-risk populations who are living in endemic regions [18]. Treatment should be applied once or twice a year if the prevalence of STH infection in the community is more than either 20% or 50%, respectively, in order to reduce the burden of morbidity induced by STH infection.

In December 2018, treatment for schistosomiasis and soil-transmitted helminthiases showed encouraging trends towards the goal of attaining a minimum target of treating at least 75% of school-age children in areas endemic for these parasitic infections to achieve the global targets set for 2020 [19].

A number of mathematical models have been employed to examine the transmission dynamics and control by mass drug administration (MDA) of soil-transmitted helminthiasis. The first of these was a deterministic structure defined by Anderson in 1980 [20], which has subsequentially been expanded to examine a variety of factors including control by mass drug administration [21–26]. These models have been used to define treatment coverage criteria for transmission elimination. They also delineate parameters that define the rate of parasite transmission and facilitate the prediction of the impact and efficiency of various control measures. Davis *et al.* [27] investigated a model for the infection of *Ascaris lumbricoides* in a human population, which incorporated the variability of egg output into the environment and the impact of both rainfall and temperature. They found that seasonal variation affects the maturation, death and transmission rates of *Ascaris lumbricoides*. Moreover, they suggested that by making

full use of seasonal variation in egg survival and maturation of *Ascaris lumbricoides*, the impact of MDA can be maximized.

Coffeng *et al*. [28] employed two mathematical models—an age-structured model and an individual-based stochastic model—to compare the transmission dynamics of *Ascaris lumbricoides* and hookworm infections and the impact of MDA with the data collected from various countries. Farrell *et al*. [29] included an age-structured deterministic model defined by a set of partial differential equations and a stochastic individual-based model built on this deterministic framework. These two models performed well in predicting the short-term impact of MDA control on *Ascaris lumbricoides* and hookworm infections. Moreover, the predictions of the models exhibited qualitatively good agreement on the impact of semi-annual versus annual MDA programmes on the entire population versus only treating school-age children.

Cooper and Hollingsworth [30] used a simple prevalence-based model to explore the impact of seasonality on the transmission dynamics and the effectiveness of an MDA programme in controlling STH *Ascaris lumbricoides* infection. They suggested that an annual MDA programme induces a greater impact in suppressing the transmission of *Ascaris lumbricoides* if it is carried out during the months with the highest temperature. They also predicted that local elimination in the community is possible if multiple annual treatments were executed at or around the estimated optimal treatment time each year. Chong *et al*. [31] used impulsive differential equations to investigate the impact of annual and biannual MDA on the mean number of worms in both treated and untreated human subpopulations. These models were employed to study community-based MDA in Kenya, showing that the interruption of transmission is likely if the efficacy of drug is sufficiently high but that interruption could happen with weaker drug efficacy and an additional round of MDA.

Here we develop a prevalence-based deterministic model to provide some general analytical insights into the transmission dynamics of STH infection in a human population. We analytically determine local stability criteria for all equilibria of the model and find bifurcation points. We add stochastic perturbation in order to examine the potential for disease extinction even when initial values are large, for sufficient bounds of the reproduction number and clumping parameter.

A prevalence-based model has some key advantages, given that most epidemiological studies and monitoring and evaluation programmes only measure this epidemiological statistic as a by-product. The more complex models that record changes in the mean-worm burden have the disadvantage that this measure is very difficult to monitor in endemic regions even when faecal output is collected in order to assess egg output by the adult worms in the human host. Egg counts have high variability and are known to be a poor measure of the true worm burden in an individual. A further reason for pursuing the properties of a prevalence-only model is to attempt to derive a deeper analytical understanding of the transmission dynamics of STH human parasites, the possible equilibrium states and dynamical behaviour around these states. Such an understanding adds to our ability to predict the impact of control programmes that repeatedly treat individuals in communities with endemic infection. Repeated treatment is required, since infection with these macroparasites do not induce protective immunity to reinfection [32].

Our prevalence-based model employs a modified form of the original deterministic equations of Anderson and May [20, 23]. We focus on investigating the prevalence of infection when no interventions are in place, in order to study the existence and stability of the possible equilibrium states employing both analytical approximations (motivated by the success in earlier work [33] using this approach) and numerical methods. We also investigate various special cases of the model in order to explore the impact of mating functions and density dependence in situations where worm burdens per host are highly aggregated and where no density-

dependent processes operate. Moreover, by adding stochastic noise into the prevalence-based model, we investigate how chance effects influence the dynamics and identify sufficient conditions for the eradication of transmission and the subsequent extinction of the parasite population.

## Mathematical model

Anderson and May (1991) [23] proposed a deterministic model based on two nonlinear differential equations to describe changes over time in the mean number of worms ($M(t)$) in a human population of density $N$ and the mean number of infectious larvae or eggs in the habitat of the human host ($L(t)$) at time $t$. In the absence of age structure, the equations are as follows:

$$
\begin{aligned}
\frac{dM}{dt} &= \beta L - \mu M \\
\frac{dL}{dt} &= \frac{\lambda}{2}\phi(M;k,z)f(M;k,z)M - \mu_0 L \equiv \frac{\lambda}{2}\mathcal{F}(M;k,z)M - \mu_0 L\,,
\end{aligned}
\tag{1}
$$

where the parameters are as defined in Table 1.

Given a negative binomial probability distribution of worm numbers per host (as observed in all epidemiological studies that have employed worm-expulsion methods), the effect of density dependence in adult worm fecundity [34] (with reductions in per capita egg output as worm density in a host rises [35]) can be described by the function

$$
f(M;k,z) = \left[1 + \frac{(1-z)M}{k}\right]^{-(k+1)}.
$$

The mating probability [36] of the adult worm is defined as

$$
\phi(M;k,z) = 1 - \left[\frac{1 + \dfrac{(1-z)M}{k}}{1 + \dfrac{(2-z)M}{2k}}\right]^{k+1},
$$

where $\mathcal{F}(M;k,z) \equiv \phi(M;k,z)f(M;k,z)$, $\gamma$ is the strength of density-dependent effects on fecundity and $z = e^{-\gamma}$. The prevalence of infection $y$ (assuming a negative binomial distribution of worms per host) is given by

$$
y = 1 - \left(1 + \frac{M}{k}\right)^{-k},
\tag{2}
$$

Table 1. Description of the associated parameters in model (1).

| Parameter | Description |
| --- | --- |
| $\beta$ | The contact rate between humans and the reservoir |
| $\mu_0$ | The per capita parasite mortality rate |
| $\mu$ | The per capita worm death rate |
| $\lambda$ | The rate of egg production per capita by female worms within a host |
| $k$ | The clumping parameter of the negative binomial distribution |
| $\gamma$ | The strength of density-dependent effects on fecundity |
| $z$ | $e^{-\gamma}$ |

where $k$ is the clumping parameter of the negative binomial distribution, which varies inversely with the degree of worm clumping.

To simplify the analysis, Anderson & May [23] considered the mean number of worms in a human population of density $N$ over time at the equilibrium of infectious larvae or eggs in the human habitat. The justification of this assumption is that the lifespan of the adult worm in the human host (1–2 years) is much longer than the life expectancy of larvae or eggs in the human host habitat (about one month or less) and, as a result, the dynamics of $L(t)$ are relatively fast compared to $M(t)$. Hence the dynamics of $M(t)$ can be redefined as

$$\frac{dM}{dt} = \mu[R_0 \mathcal{F}(M; k, z) - 1]M, \tag{3}$$

where $R_0 = \beta\lambda/(2\mu_0\mu)$ is the basic reproduction number for the parasite in the absence of density-dependence in adult worm fecundity [23].

By considering a situation at equilibrium where infection occurs continuously in the human host population with a constant force of infection as in (2) and where no intervention/control strategy has been implemented, the transmission dynamics as previously measured by the mean-worm burden of STH within the human population can be converted into a prevalence of infection (given a negative binomial burden of worms per host with fixed $k$) as described in the following set of equations:

$$
\begin{aligned}
\frac{dx}{dt} &= -\beta x^{\frac{k+1}{k}} L + \mu\left[\frac{\mathcal{P}(1; M, k)}{1 - \mathcal{P}(0; M, k)}\right] y = (-\beta L + \mu k W_1)x^{\frac{k+1}{k}} \\
\frac{dy}{dt} &= \beta x^{\frac{k+1}{k}} L - \mu\left[\frac{\mathcal{P}(1; M, k)}{1 - \mathcal{P}(0; M, k)}\right] y = (\beta L - \mu k W_1)x^{\frac{k+1}{k}} \\
\frac{dL}{dt} &= \frac{1}{2}\lambda k W_1 \mathcal{F}(y; k, z) - \mu_0 L,
\end{aligned}
\tag{4}
$$

subject to the restriction $x + y = 1$ and having redefined $\mathcal{F}(y; k, z) = \phi(y; k, z)f(y; k, z)$ and $y(t) = Y(t)/N(t)$, where $W_1 = (1 - y)^{-\frac{1}{k}} - 1$, $N(t) = X(t) + Y(t)$ is the total population of susceptible $X(t)$ and infected $Y(t)$ humans at time $t$ and $x(t) = X(t)/N(t)$. The description for each parameter of model (4) is as defined in Table 1. Note that the first two equations may also be derived by substituting Eq (2) into Eq (1) for a change of variables.

An infected individual who has a worm burden of one parasite has a high probability to recover (even without treatment) due to the death of worm and the host moving to the susceptible uninfected class. Thus the probability that an infected individual has only one worm is given by $\mathcal{P}(1; M, k)/[1 - \mathcal{P}(0; M, k)]$. This is the conditional probability of being in the $w = 1$ class given the probability of having at least $w = 1$ worm. That is,

$$\mathcal{P}(w = 1 | w \geq 1) = \frac{\mathcal{P}(w = 1 \wedge w \geq 1)}{\mathcal{P}(w \geq 1)} = \frac{\mathcal{P}(w = 1)}{1 - \mathcal{P}(w = 0)},$$

where the probability of an infected individual getting $w$ worms is

$$\mathcal{P}(w; M, k) = \frac{\Gamma(k + w)}{\Gamma(k)w!}\left(\frac{k + M}{k}\right)^{-k}\left(\frac{M + k}{M}\right)^{-w}.$$

Both $f(M; k, z)$ and $\phi(M; k, z)$ can be rewritten in terms of prevalence $y$, clumping $k$ and fecundity $z$ as follows:

$$
\begin{aligned}
f(y; k, z) &= \{1 + (1 - z)[(1 - y)^{-1/k} - 1]\}^{-(k+1)} \\
\phi(y; k, z) &= 1 - \left\{ \frac{1 + (1 - z)[(1 - y)^{-1/k} - 1]}{1 + \frac{1}{2}(2 - z)[(1 - y)^{-1/k} - 1]} \right\}^{k+1}.
\end{aligned}
$$

We can therefore rewrite model (4) given the equilibrium state of $L$ using the substitution $x = 1 - y$ as follows:

$$
\frac{dy}{dt} = \mu k W_1 (1 - y)^{\frac{k+1}{k}} [R_0 \mathcal{F}(y; k, z) - 1]. \tag{5}
$$

Throughout this work, we choose $z = 0.96$ and $\mu = 0.5$ per year, unless otherwise stated, given published values of these parameters [23]. Before we further analyse the prevalence model (5), we would like to show the comparison of prevalence values generated by models (3) and (5) in Fig 1. We observe that both models have good agreements for arbitrary $k$ and initial values, so we focus our attention on the simpler prevalence model (5) for the study of STH infection in human populations.

It is clear that $y = 0$ and $y = 1$ are equilibrium points of model (5). The other endemic equilibrium, $y^*$, of model (5) exists whenever we solve $R_0 \mathcal{F}(y^*; k, z) = 1$. By solving $R_0 \mathcal{F}(y^*; k, z) = 1$ numerically (using the bisection method), we are able to depict the solutions of $y^*$ with different $k$ values in Fig 2. From this figure, we can see that there are two distinct equilibrium solutions. One tends to zero (elimination of transmission) and another one tends to a value of endemic infection. These two states are separated by an unstable equilibrium commonly termed the 'transmission breakpoint'. We denote a stable equilibrium solution by the solid curve, whereas the unstable equilibrium solution is represented by the dashed curve. By increasing the $R_0$ values, the stable equilibrium solutions get larger, but the unstable equilibrium solutions get smaller. In addition, there is a bifurcation point (denoted by $y_{\mathrm{bp}}$) in model

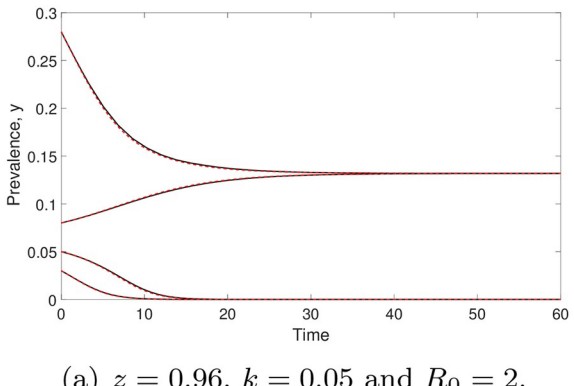

(a) $z = 0.96$, $k = 0.05$ and $R_0 = 2$.

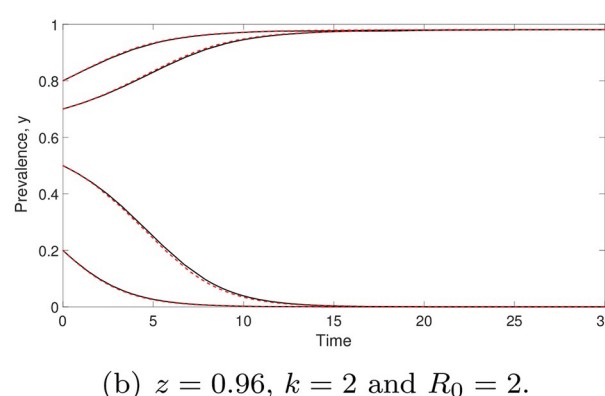

(b) $z = 0.96$, $k = 2$ and $R_0 = 2$.

**Fig 1. Comparisons of prevalence values generated by the models (3) (red dashed line) and (5) (black solid curve).** Both models (prevalence-based or mean-worm-burden-based) produce well-matched results for arbitrary $k$ and initial values. Both predict that the infection will either die off or reach an endemic state.

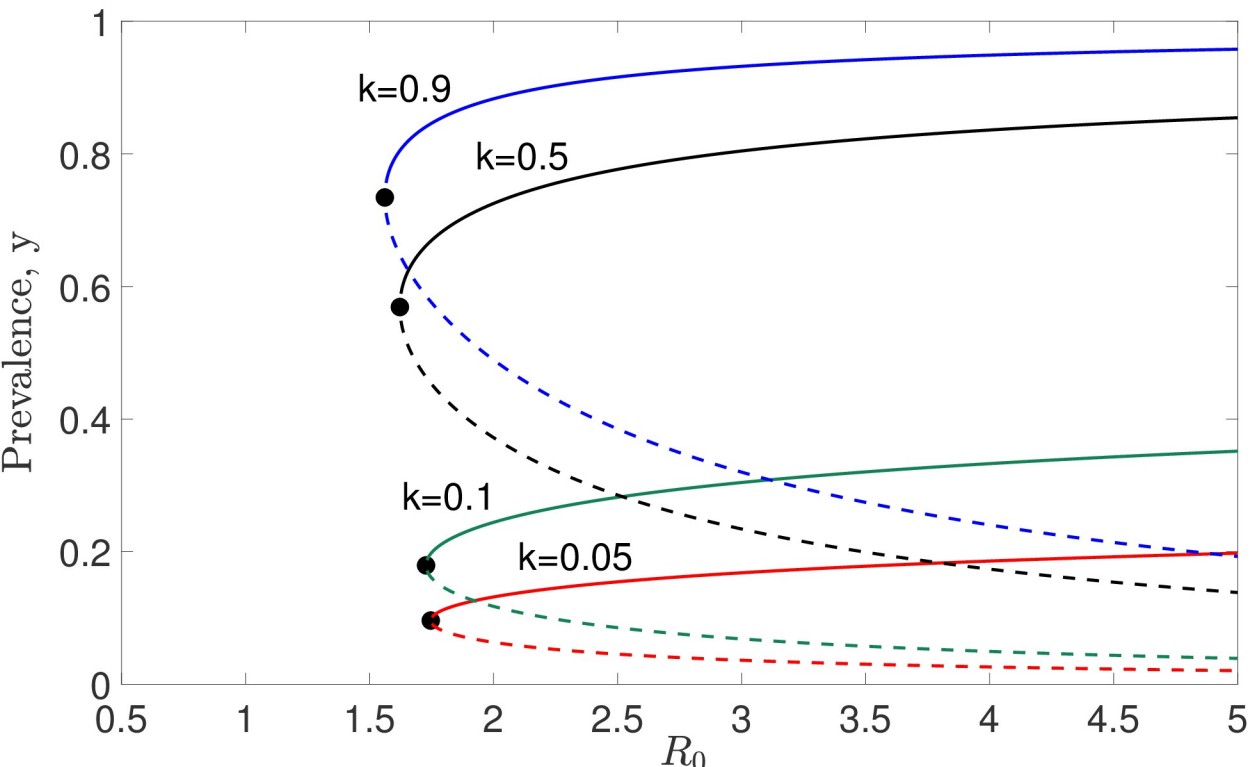

**Fig 2. Numerical solutions of equilibrium $y^*$ as a function of $R_0$ with different $k$ values, but fixed $z = 0.96$.**

(5), given by

$$y_{\text{bp}} = 1 - \left( \frac{\left[\frac{2-z}{2(1-z)}\right]^{-\left(\frac{k+1}{k+2}\right)} - 1}{\frac{z}{2-z}\left\{1 - 2\left[\frac{2-z}{2(1-z)}\right]^{\frac{1}{k+2}}\right\}} \right)^k .$$

See the S1 Appendix for further details.

**Theorem 1** *Let* $W_1^* = (1-y^*)^{-\frac{1}{k}} - 1$, $\mathcal{F}'(y;k,z) = d\mathcal{F}(y;k,z)/dy$ *and*

$$\mathcal{F}'(y^*;k,z) = \frac{(k+1)(2-z)(1-y^*)^{-\left(\frac{k+1}{k}\right)}}{2k}\left[1 + \frac{(2-z)W_1^*}{2}\right]^{-(k+2)}$$

$$- \frac{1}{k}(k+1)(1-z)(1-y^*)^{-\left(\frac{k+1}{k}\right)}\left[1 + (1-z)W_1^*\right]^{-(k+2)} .$$

*Model* (5) *always achieves local asymptotic stability in the absence of infection (i.e., $y = 0$), whereas it is unstable if $y = 1$. In addition, if $y^* > (<)y_{\text{bp}}$, the endemic equilibrium ($y^*$) of model (5) is locally asymptotically stable (unstable). A local fold bifurcation occurs at $y_{\text{bp}}$.*

**Proof**. *Let* $\hat{\lambda}$ *represent the eigenvalue of model* (5). *The eigenvalue of model* (5) *is defined as follows*:

$$\hat{\lambda} = \mu\left[(k+1)(1-y)^{\frac{1}{k}} - k\right]\left[R_0\mathcal{F}(y;k,z) - 1\right]$$
$$+ \mu k R_0 W_1(1-y)^{\frac{k+1}{k}} \mathcal{F}'(y;k,z).$$
(6)

In the absence of infection, $\hat{\lambda} = -\mu < 0$ since $\mu > 0$. This proves that $y = 0$ is locally asymptotically stable. For $y = 1$, $\hat{\lambda} = \mu k > 0$ since $\mu, k > 0$. Hence $y = 1$ is an unstable equilibrium point. Moreover, for $y = y^*$,

$$\hat{\lambda}|_{y=y^*} = \mu k R_0 W_1^*(1-y^*)^{\frac{k+1}{k}} \mathcal{F}'(y^*;k,z).$$

$\hat{\lambda}|_{y=y^*} < (>)0$ is equivalent to $\mathcal{F}'(y^*;k,z) < (>)0 \Rightarrow y^* > (<)y_{\text{bp}}$. As a result, $\hat{\lambda}|_{y=y^*} < (>)0$ whenever $y^* > (<)y_{\text{bp}}$. Hence $y^*$ is locally asymptotically stable (unstable) if $y^* > (<)y_{\text{bp}}$. When $y = y_{\text{bp}}$, we obtain $\mathcal{F}'(y_{\text{bp}};k,z) = 0$; hence $\hat{\lambda}|_{y=y_{\text{bp}}} = 0$. Therefore a local bifurcation occurs at $y_{\text{bp}}$.

Denote the unstable and stable endemic equilibria of model (5) as $y_*$ and $y^*$, respectively. By varying the $R_0$ values (as in Fig 2), we describe the relationship between the eigenvalue (6) and the endemic equilibrium of model (5) in Fig 3 with different $k$ values. By increasing the $y$ values, the eigenvalue, $\hat{\lambda}$, changes from positive to negative values for fixed $k$ values. This implies that the stability of model (5) changes from unstable (dashed curve) to stable (solid curve). A local fold bifurcation occurs when $\hat{\lambda} = 0$. In other words, the local fold bifurcation occurs at the point where the signs of eigenvalue and the stability of an equilibrium point are changing.

The dynamics of model (5) are depicted in Fig 4. We represent the stable and unstable equilibria by filled and unfilled circles, respectively. For arbitrary $k$ and $R_0$, all solutions of model (5) tend to zero as $t \to \infty$ if the initial prevalence value satisfies $y_0 < y_*$. However, the solutions of model (5) will approach a stable endemic equilibrium ($y^*$) whenever $y_0 > y_*$. Since there is an unstable equilibrium point ($y_*$) existing in the model, a separatrix between the $\omega$-limit sets of these two stable equilibria ($y = 0$ and $y^*$) is formed. Based on the numerical results in this figure, we can summarize that, for any initial prevalence value lower than $y_*$, model (5) predicts disease extinction. Otherwise, this model forecasts that parasite infection will remain in the endemic state. Note that $y^*$ increases whenever $k$ and $R_0$ values are increasing. In order to better describe the dynamics of model (5), we depict the vector field of $y$ in Fig 5.

In Fig 6, we illustrate the heatmaps of the vector field of $y$ (i.e., Eq (5)) and the second derivative of $y$ with respect to time:

$$\frac{d^2y}{dt^2} = \mu\left[(k+1)(1-y)^{\frac{1}{k}} - k\right]\left[R_0\mathcal{F}(y;k,z) - 1\right]$$
$$+ \mu k R_0 W_1(1-y)^{\frac{k+1}{k}} \mathcal{F}'(y;k,z).$$
(7)

The vector field depicts the velocity of $y$, whereas the second derivative with respect to time describes the acceleration of $y$. The positive (negative) value of Eq (5) in this figure

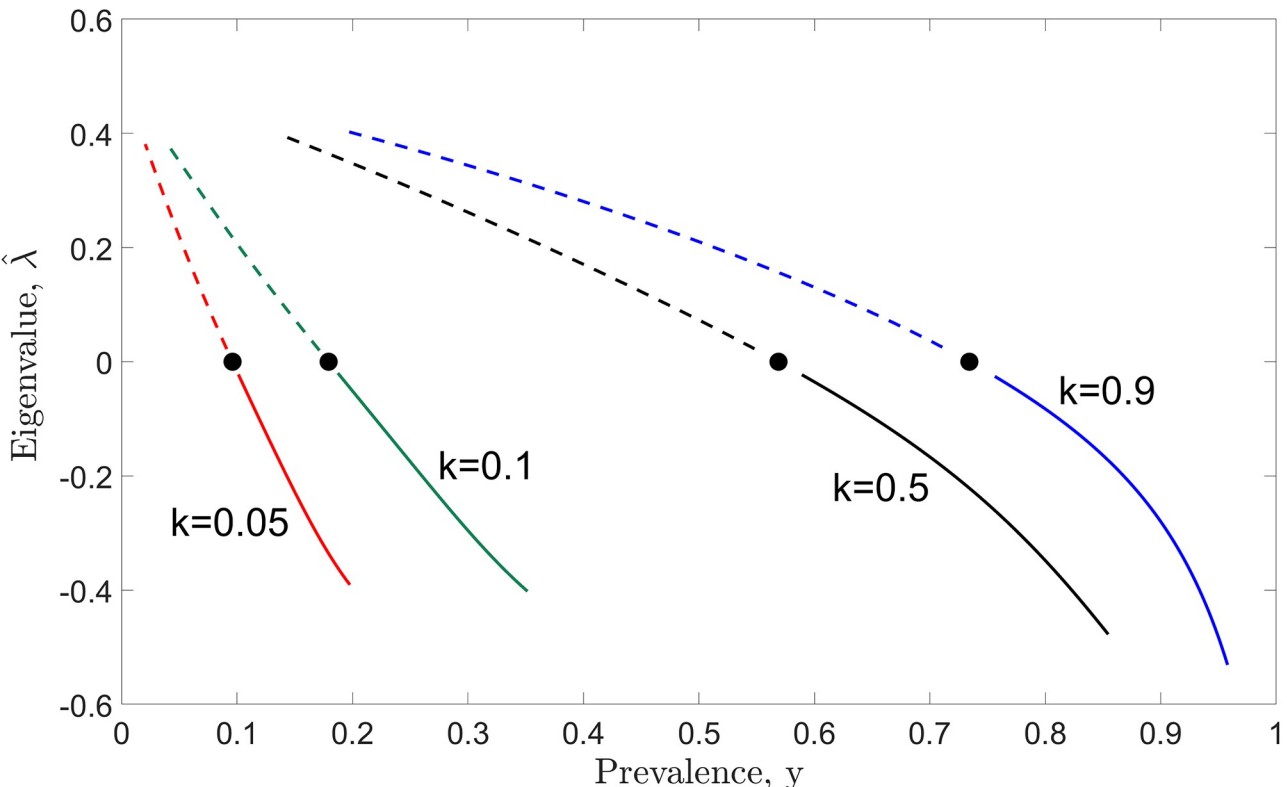

**Fig 3. The relationship between the eigenvalue ([6]) and the endemic equilibrium of the model ([5]) is demonstrated by varying $k$ values (corresponding to $0 \leq R_0 \leq 5$).** Linearization is one of the key methods employed in assessing stability, and it can be applied to determine the local stability of a model governed by ordinary differential equations. By definition [37], an equilibrium point is locally asymptotically stable if all eigenvalues have negative real parts, whereas it is unstable if at least one eigenvalue has positive real part. A local bifurcation occurs whenever the real part of an eigenvalue passes through zero.

corresponds to the force acting in an upward (downward) direction. Thus, if the trajectory/ solution of model ([5]) is moving upward (downward) quickly, then ([5]) will have a large positive (negative) value. Eq ([7]) measures how "fast" ([5]) changes with respect to time $t$, which is illustrated in Fig 6(b). The movement of the solution $y$ slows down (speeds up) if the velocity and acceleration of $y$—i.e., Eqs ([5]) and ([7]), respectively—have opposite (identical) signs.

In Fig 6(a), the rate of change of $y$ at the equilibrium points is equal to zero; i.e., $y = 0$, the stable equilibrium point $y^*$ (denoted by the solid black curve), the unstable equilibrium point $y_*$ (represented by the dashed black curve) and the point $y = 1$. The velocity of $y$ is moving towards $y = 0$ and $y^*$. However, it is moving away from $y_*$ and $y = 1$. In addition, for $y_* < y < y^*$, the trajectory of model ([5]) around the transmission breakpoint ($y_*$) moves quickly from $y_*$ and converges to the stable equilibrium point ($y^*$) at a slower pace. For $y > y^*$, the trajectory of the model ([5]) is speeding up and moving faster towards $y^*$. For $y < y_*$, the trajectory of model ([5]) moves slowly towards the state of parasite extinction once it has crossed the transmission breakpoint. However, its movement speeds up when it is close enough to $y = 0$. This reveals that both $y = 0$ and $y^*$ are stable equilibria, whereas $y_*$ and $y = 1$ are unstable equilibria, validating Theorem 1.

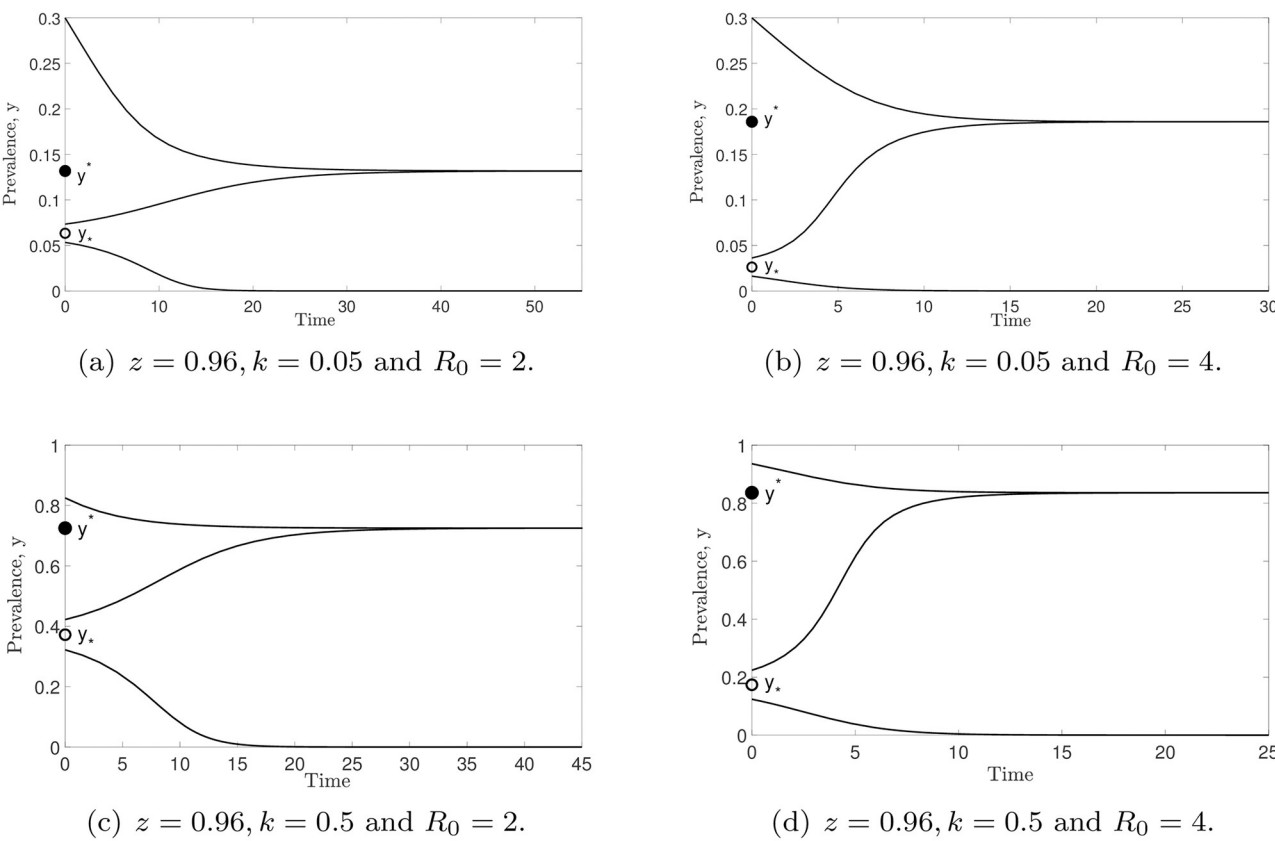

(a) $z = 0.96, k = 0.05$ and $R_0 = 2$.

(b) $z = 0.96, k = 0.05$ and $R_0 = 4$.

(c) $z = 0.96, k = 0.5$ and $R_0 = 2$.

(d) $z = 0.96, k = 0.5$ and $R_0 = 4$.

**Fig 4. The dynamics of model (5) when varying $k$, $R_0$ and initial value $y_0$.** By varying $k$ and $R_0$ values, all solutions of this model converge to zero if $y_0$ < $y_*$, whereas the solutions of this model approach the endemic equilibrium $y^*$ whenever $y_0 > y_*$ as $t \to \infty$.

## Analytical approximations around the equilibrium points

In this section, we examine the behaviour of model (5) around the equilibrium points. This has practical relevance in terms of interpreting trends in the field as control measures intensify and moving the system toward the unstable equilibrium point and away from the stable endemic state. We shall expand model (5) up to $\mathcal{O}(y^2)$ around the equilibrium points and then

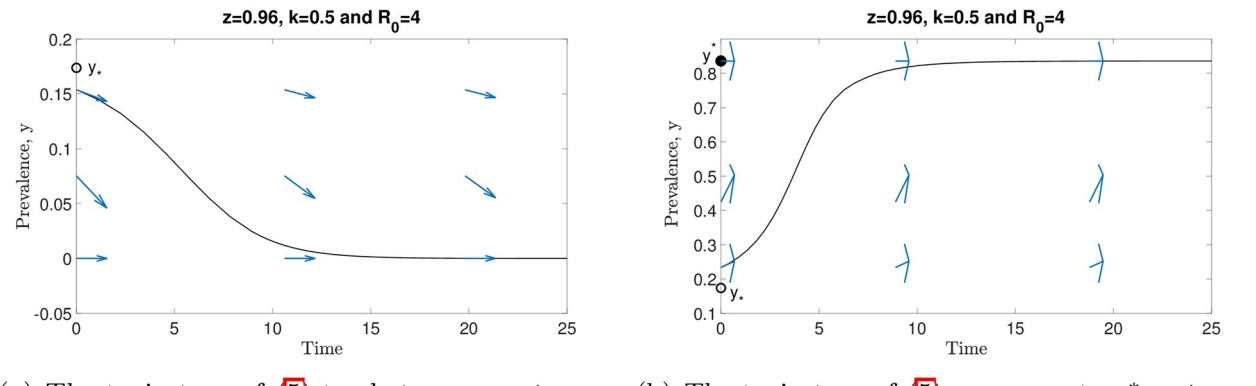

(a) The trajectory of (5) tends to zero as $t \to \infty$. (b) The trajectory of (5) converges to $y^*$ as $t \to \infty$.

**Fig 5. The vector field (5) derived using numerical solutions of the model (5), where $z = 0.96$, $k = 0.5$ and $R_0 = 4$.**

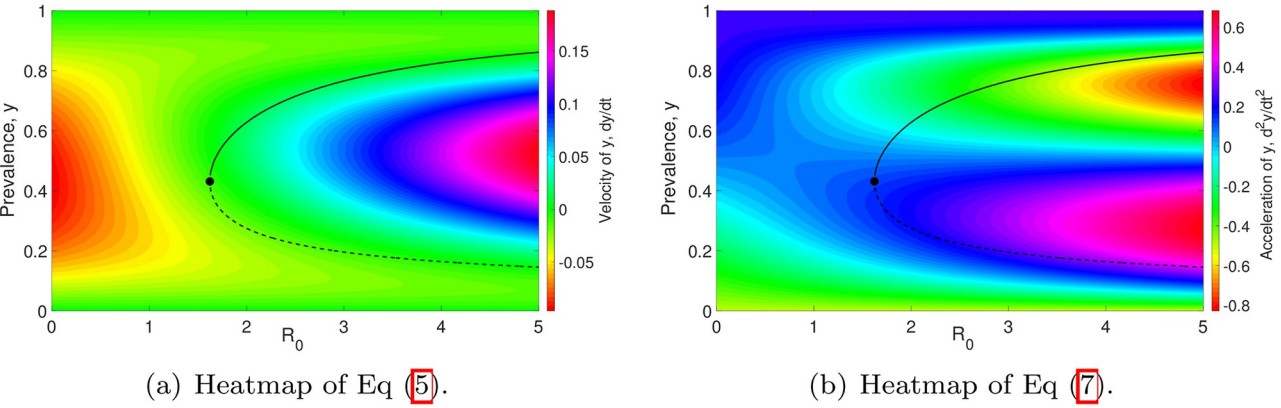

**Fig 6. Heatmaps of the vector field (5) and the rate of change of the model (5)—i.e., Eq (7)—are as shown in subfigures (a) and (b), respectively.** $k = 0.5$ and $z = 0.96$ are chosen to generate these two plots.

look for the corresponding analytical solution. In addition, the analytical and numerical solutions of the model (5) will be compared. Recall that the Taylor series expansion of a function about a point $b$ takes the following general form:

$$g(y) = g(b) + g'(b)(y - b) + \frac{g''(b)(y - b)^2}{2!} + \frac{g'''(b)(y - b)^3}{3!} + \mathcal{O}(y^4).$$

## Analytical approximation around $y = 0$

As $y \to 0$, we have $(1 - y)^{-\frac{1}{k}} \approx 1 + y/k$, $(1 - y)^{\frac{k+1}{k}} \approx 1 - y[(k + 1)/k]$ and

$$\mathcal{F}(y; k, z) \approx \frac{z(k + 1)}{2k} y \left[ 1 + \frac{(k + 2)(3z - 4)}{4k} y \right]$$

$$+ \frac{z(k + 1)(k + 2)(k + 3)(7z^2 - 18z + 12)}{48k^3} y^3.$$

Hence, up to $\mathcal{O}(y^2)$, the expansion of model (5) around $y = 0$ is given by

$$\frac{dy}{dt} = \mu \left[ \frac{(k + 1)(R_0 z + 2)}{2k} y - 1 \right] y, \tag{8}$$

and the analytical solution of (8) is

$$y(t) = \frac{2ky_0}{(k + 1)(R_0 z + 2)(1 - e^{\mu t})y_0 + 2ke^{\mu t}}, \tag{9}$$

where $y_0$ is the initial value of $y$. The expansion (8) has a good agreement with the full solution if

$$y < \min \left\{ \frac{6k}{k + 3} \left| \frac{3z - 4}{7z^2 - 18z + 12} \right|, \left| \frac{3k}{k - 1} \right|, \frac{3k}{2k + 1} \right\} \equiv y_{agr}, \tag{10}$$

where $k \neq 1$ and $7z^2 - 18z + 12 \neq 0$.

Fig 7 demonstrates both numerical and analytical (with approximations) solutions of model (5) when $y \to 0$. We can see that by increasing the $R_0$ value, $y_*$ is decreasing. However, if

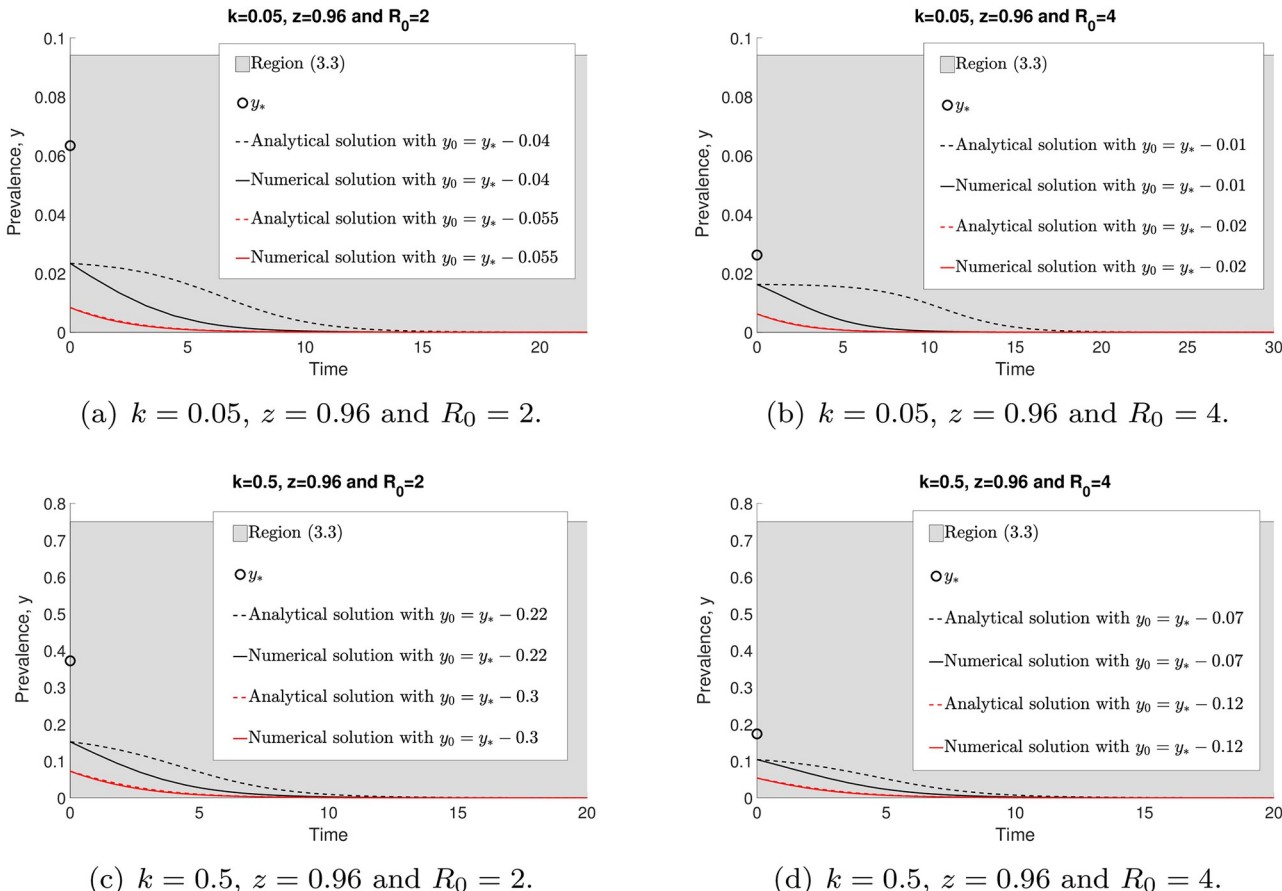

**Fig 7. The comparisons of the analytical (9) and numerical (with approximations) (5) solutions with arbitrary initial points around $y = 0$ and $y_0 < y_*$.**

$k$ is increasing, both region (10) (in grey) and $y_*$ are increasing. Both analytical (9) and numerical (5) solutions have closer agreement when the initial value $y_0$ is approaching zero. Nevertheless, both of these solutions with arbitrary $y_0 < y_*$ converge to zero as $t \to \infty$.

## Analytical approximation around equilibrium point

Let $y_{\mathrm{eq}}$ denote the nontrivial equilibrium point of model (5). Up to $\mathcal{O}(y^2)$, the expansion of model (5) around $y_{\mathrm{eq}}$ is

$$\frac{dy}{dt} = \mu k R_0 \left( \hat{A}_0 y^2 + \hat{B}_0 y + \hat{C}_0 \right), \tag{11}$$

where

$$\hat{A}_0 = \left[ \frac{(k+1)(1-y_{\text{eq}})^{\frac{1}{k}}}{k} - 1 \right] \mathcal{F}'(y_{\text{eq}};k,z)$$

$$+ \frac{W_1^{\text{eq}}}{2}(1-y_{\text{eq}})^{\frac{k+1}{k}} \mathcal{F}''(y_{\text{eq}};k,z),$$

$$\hat{B}_0 = W_1^{\text{eq}}(1-y_{\text{eq}})^{\frac{k+1}{k}} [\mathcal{F}'(y_{\text{eq}};k,z) - y_{\text{eq}}\mathcal{F}''(y_{\text{eq}};k,z)]$$

$$+ 2y_{\text{eq}} \left[ 1 - \frac{(k+1)(1-y_{\text{eq}})^{\frac{1}{k}}}{k} \right] \mathcal{F}'(y_{\text{eq}};k,z),$$

$$\hat{C}_0 = \left[ \frac{(k+1)(1-y_{\text{eq}})^{\frac{1}{k}}}{k} - 1 \right] \mathcal{F}'(y_{\text{eq}};k,z)y_{\text{eq}}^2$$

$$+ W_1^{\text{eq}}(1-y_{\text{eq}})^{\frac{k+1}{k}} \left[ \frac{y_{\text{eq}}\mathcal{F}''(y_{\text{eq}};k,z)}{2} - \mathcal{F}'(y_{\text{eq}};k,z) \right] y_{\text{eq}}$$

$$W_1^{\text{eq}} = (1-y_{\text{eq}})^{\frac{1}{k}} - 1.$$

The analytical solution of (11) is given by

$$y(t) = y_{\text{eq}} - \frac{\hat{A}\left\{ 1 - \tanh\left[ -\frac{\mu k R_0 \hat{A} t}{2} + \tanh^{-1}\left( 1 + (y_0 - y_{\text{eq}})\hat{B} \right) \right] \right\}}{2 \left[ \frac{(k+1)(1-y_{\text{eq}})^{\frac{1}{k}} - k}{k} \right] \mathcal{F}'(y_{\text{eq}};k,z) + W_1^{\text{eq}}(1-y_{\text{eq}})^{\frac{k+1}{k}} \mathcal{F}''(y_{\text{eq}};k,z)}, \tag{12}$$

where

$$\hat{A} = W_1^{\text{eq}}(1-y_{\text{eq}})^{\frac{k+1}{k}} \mathcal{F}'(y_{\text{eq}};k,z),$$

$$\hat{B} = \frac{2 \left[ \frac{(k+1)(1-y_{\text{eq}})^{\frac{1}{k}} - k}{k} \right] \mathcal{F}'(y_{\text{eq}};k,z) + W_1^{\text{eq}}(1-y_{\text{eq}})^{\frac{k+1}{k}} \mathcal{F}''(y_{\text{eq}};k,z)}{\hat{A}}.$$

This analytical solution is only defined if

$$
\mathcal{F}''(y_{\mathrm{eq}}; k, z) \neq \frac{2\left[1 - \frac{(k+1)(1 - y_{\mathrm{eq}})^{\frac{1}{k}}}{k}\right]\mathcal{F}'(y_{\mathrm{eq}}; k, z)}{W_1^{\mathrm{eq}}(1 - y_{\mathrm{eq}})^{\frac{k+1}{k}}}
$$

and $\mathcal{F}'(y_{\mathrm{eq}}; k, z) > 0$. Furthermore, due to the Taylor expansion, (11) is accurate if

$$
|y - y_{\mathrm{eq}}| < \min\left\{1, 3\left|\frac{W_2}{W_3}\right|\right\} \equiv y_{agr1} , \tag{13}
$$

where

$$
\begin{aligned}
W_2 ={}& W_1^{\mathrm{eq}}(1 - y_{\mathrm{eq}})^{\frac{k+1}{k}}\mathcal{F}''(y_{\mathrm{eq}}; k, z) \\[4pt]
&+ 2\left[\frac{(k+1)(1 - y_{\mathrm{eq}})^{\frac{1}{k}}}{k} - 1\right]\mathcal{F}'(y_{\mathrm{eq}}; k, z) ,
\end{aligned}
$$

$$
\begin{aligned}
W_3 ={}& W_1^{\mathrm{eq}}(1 - y_{\mathrm{eq}})^{\frac{k+1}{k}}\mathcal{F}'''(y_{\mathrm{eq}}; k, z) \\[4pt]
&+ \frac{3\left[(k+1)(1 - y_{\mathrm{eq}})^{\frac{1}{k}} - k\right]\mathcal{F}''(y_{\mathrm{eq}}; k, z)}{k} \\[4pt]
&- 3\left(\frac{k+1}{k^2}\right)(1 - y_{\mathrm{eq}})^{-\left(\frac{k-1}{k}\right)}\mathcal{F}'(y_{\mathrm{eq}}; k, z) .
\end{aligned}
$$

In Fig 8, we observe that the solutions of (5) and (12) converge to the endemic equilibrium, $y^*$, and give good agreement within the region between analytical work based on an approximation and numerical evaluations (13) (grey region), especially for initial values $y_0$ that are sufficiently close to $y^*$. Moreover, $y^*$ is increasing whenever $k$ and $R_0$ values are increasing.

We illustrate both analytical and numerical solutions (Eqs (12) and (5), respectively) around the unstable endemic equilibrium point, $y_*$, in Fig 9. All analytical and numerical solutions are moving away from $y_*$ for arbitrary $k$ and $R_0$ values as $t \to \infty$. Nevertheless, both solutions lead to a good agreement within region (13) if $y_0$ is close enough to $y_*$. Furthermore, we find that $y_*$ is increasing if $k$ is increasing. However, by increasing $R_0$, $y_*$ gets smaller. We will discuss the approximation accuracy of (5) as $k \to 0$ in the next section.

## Special cases

### Case 1: $k \to 0$

As the clumping parameter $k \to 0$, the probability distribution of STH parasites within the human host population becomes highly aggregated. It is possible to have very few individuals in the population who carry a large burden of parasites while the remainder of the population

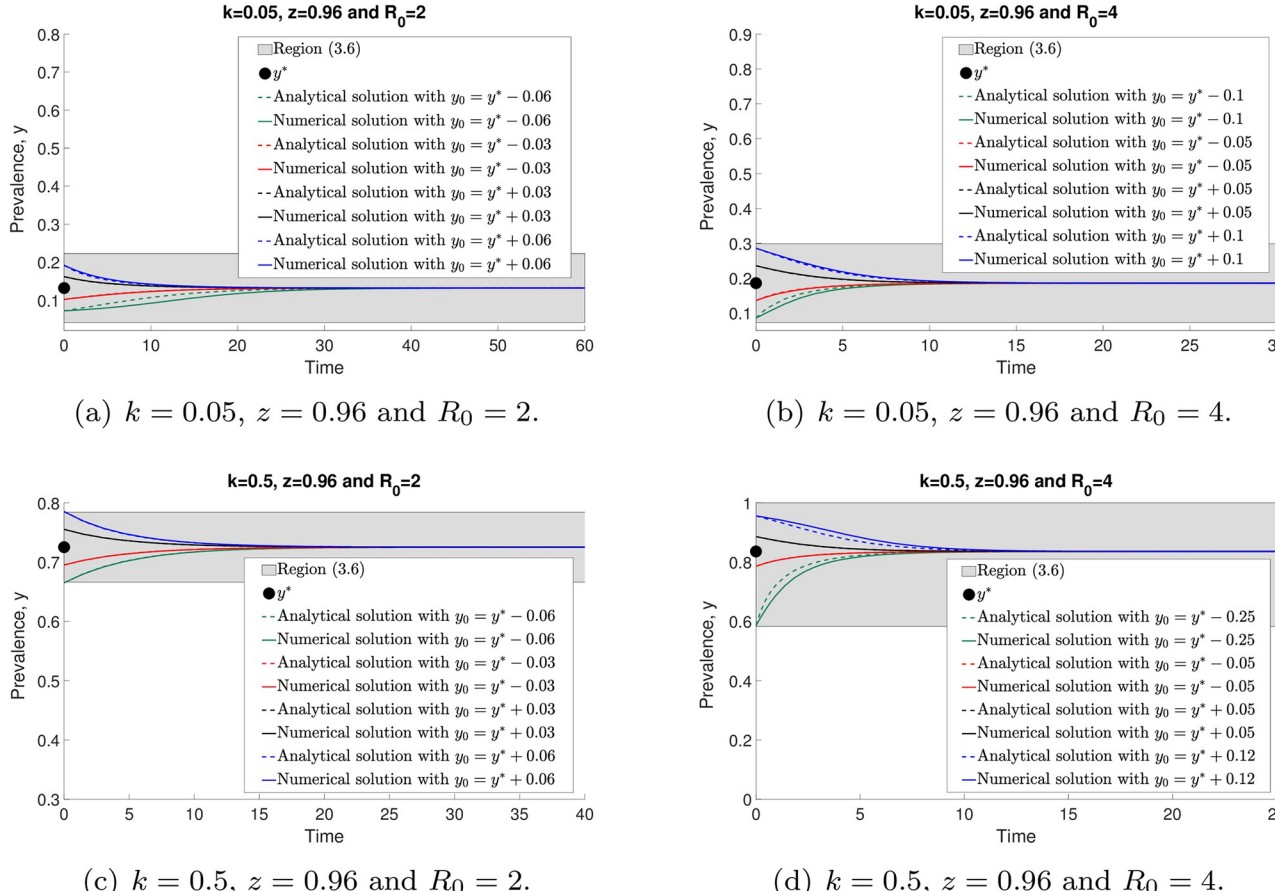

**Fig 8. The comparisons of analytical (12) and numerical (5) solutions with arbitrary initial points around the stable endemic equilibrium, $y^*$.**

has very few parasites. In the limit, all parasites are harbored by one individual host. As aggregation increases, if we manage to identify and provide appropriate treatment to those carrying worms, the prevalence of STH in the population will be reduced significantly. This improves the likelihood of transmission eradication; hence we are interested in approximating the solution of model (5) as $k \to 0$. As $k \to 0$, we obtain

$$\mathcal{F}(y;z) \approx \frac{z(1-y)^{\frac{1}{k}}}{(1-z)(2-z)} \approx 0 \quad \text{almost everywhere.} \tag{14}$$

The approximation of (14) is supported by Fig 10, where $\mathcal{F}(y;z)$ is zero almost everywhere as $k \to 0$. In addition, from this figure, we can see that the infection is concentrated in very few people. Thus the model (5) can be simplified to

$$\frac{dy}{dt} \approx -\mu k W_1 (1-y)^{\frac{k+1}{k}}, \tag{15}$$

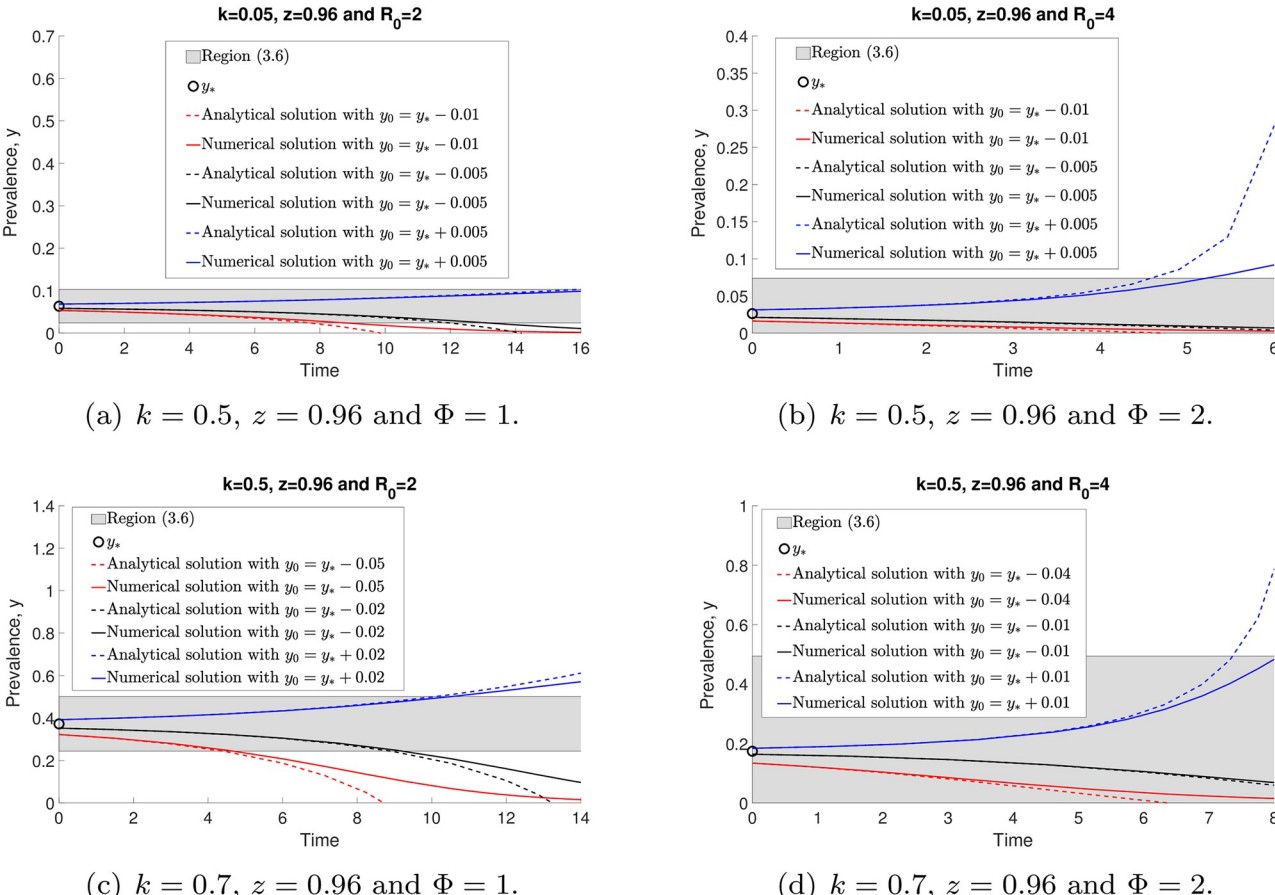

**Fig 9. The comparisons of analytical (12) and numerical (5) solutions with arbitrary initial points around the unstable endemic equilibrium, $y_*$.**

and the analytical solution of (15) is defined as

$$y(t) = 1 - \left[ \frac{(1-y_0)^{\frac{1}{k}} e^{\mu t}}{1 - (1-y_0)^{\frac{1}{k}}(1 - e^{\mu t})} \right]^k, \tag{16}$$

where $y_0$ is the initial value of $y(t)$. It is clear that $y = 0$ and $y = 1$ are the equilibria values for model (15), and the eigenvalue for model (15) is $\hat{\lambda}_k = \mu\left[k - (k+1)(1-y)^{\frac{1}{k}}\right]$. Since the rate of change of $y$ in model (15) is governed by a decreasing function of $y$, the solution will eventually approach $y = 0$ as $t \to \infty$. Moreover, at $y = 0$, the eigenvalue $\hat{\lambda}_k = -\mu < 0$, which shows that $y = 0$ is a locally asymptotically stable equilibrium point. It follows that parasite eradication is possible in this case. However, $y = 1$ is an unstable equilibrium point since $\hat{\lambda}_k|_{y=1} = \mu k > 0$.

The analytical solution (16) and numerical solution of model (5) around the stable equilibrium $y = 0$ as $k \to 0$ are illustrated in Fig 11. From this figure, we see that both analytical and numerical solutions are in good agreement and converging to zero whenever $y_0 < y_*$ for arbitrary $R_0$ and small $k$ values. Both models ((5) and (15)) predict disease extinction whenever $y_0 < y_*$.

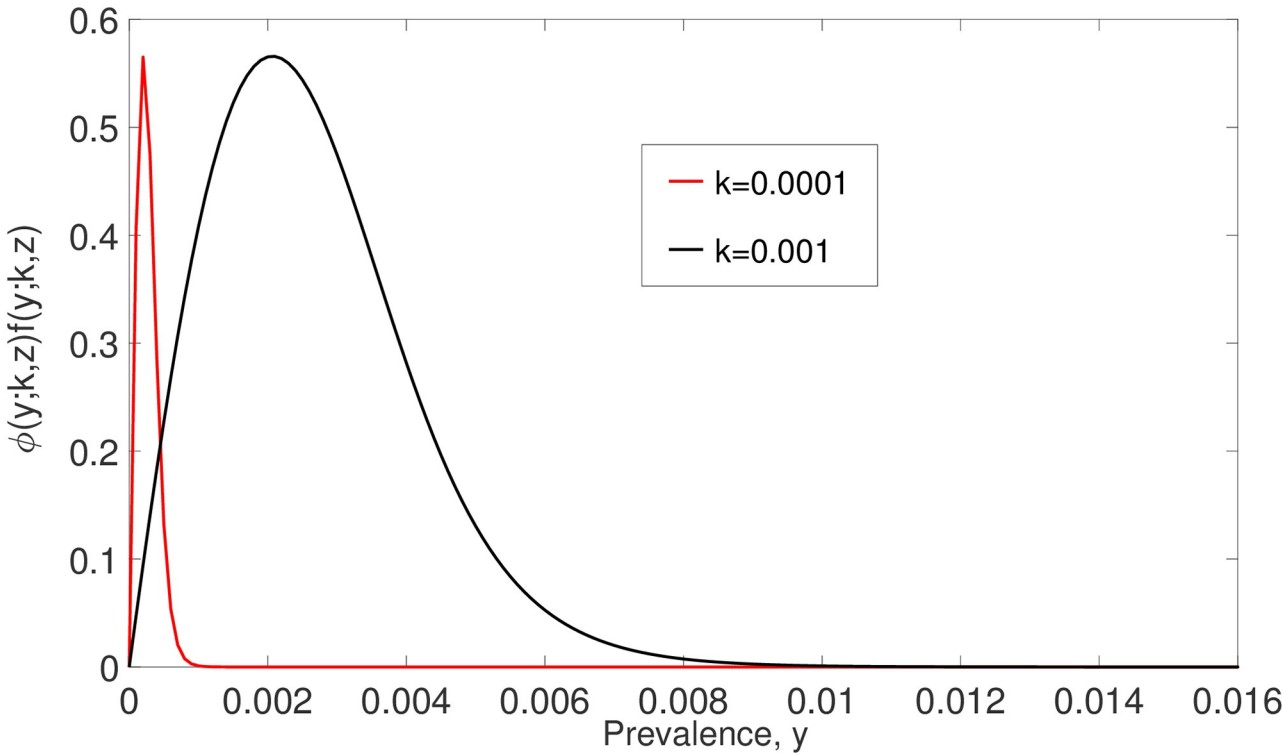

**Fig 10. The relationship between $\mathcal{F}(y; k, z)$ and $y$ by varying parameter $k$.** $\mathcal{F}(y; k, z)$ is zero almost everywhere as $k \to 0$.

### Case 2: $z = 1$

The second special case is where there are no density-dependent effects acting on the parasite's fecundity (i.e., by assuming $\gamma = 0$). When $z = 1$,

$$\mathcal{F}(y; k) = 1 - (1 + \frac{W_1}{2})^{-(k+1)}. \tag{17}$$

The relationship between $\mathcal{F}(y; k)$ and $y$ is shown in Fig 12, and we can see that $\mathcal{F}(y; k)$ is a non-decreasing function of $y$.

By substituting (17) into (5), we obtain

$$
\begin{aligned}
\frac{dy}{dt} &= \mu k W_1 (1 - y)^{\frac{k+1}{k}} [R_0 \mathcal{F}(y; k) - 1] \\
&= \mu k W_1 (1 - y)^{\frac{k+1}{k}} \left\{ R_0 \left[ 1 - \left( 1 + \frac{W_1}{2} \right)^{-(k+1)} \right] - 1 \right\}.
\end{aligned} \tag{18}
$$

Since $y = 0$ and $y = 1$ are equilibrium points for model (18), by solving $R_0 \mathcal{F}(y_z^*; k) - 1 = 0$ for $y_z^*$, we obtain

$$y_z^* = 1 - \left[ 2 \left( \frac{R_0}{R_0 - 1} \right)^{\frac{1}{k+1}} - 1 \right]^{-k},$$

which exists when $R_0 \neq 1$; $y_z^*$ is another equilibrium point for model (18).

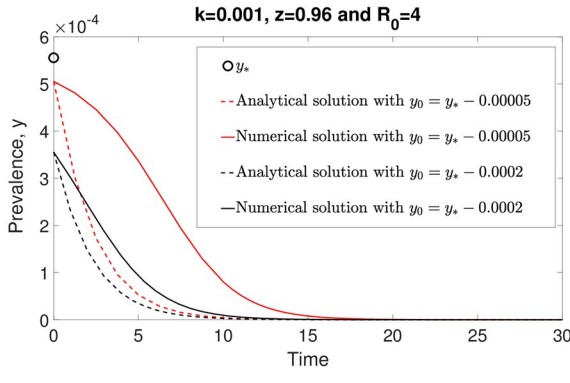

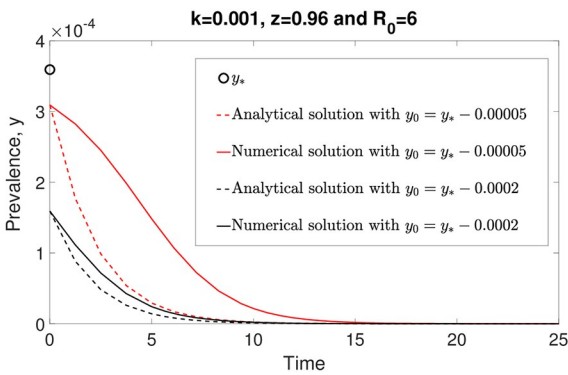

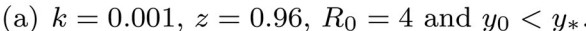

(a) $k = 0.001$, $z = 0.96$, $R_0 = 4$ and $y_0 < y_*$.

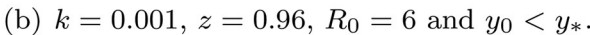

(b) $k = 0.001$, $z = 0.96$, $R_0 = 6$ and $y_0 < y_*$.

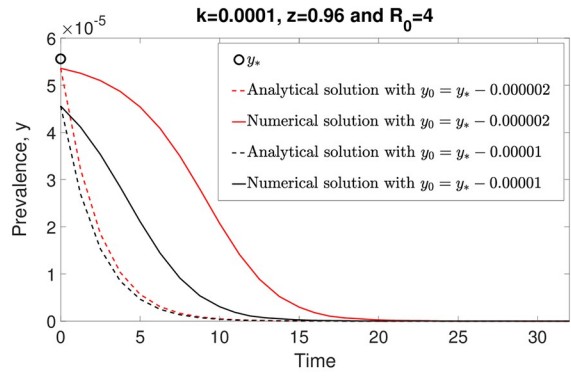

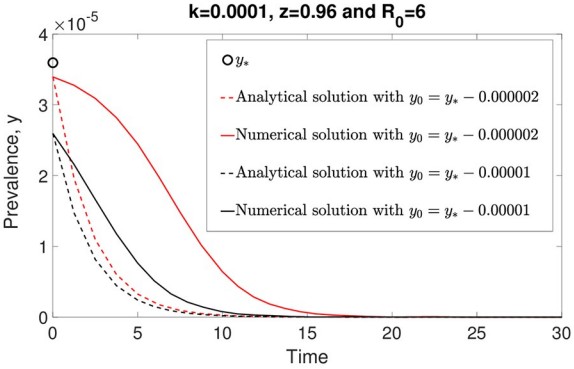

(c) $k = 0.0001$, $z = 0.96$, $R_0 = 4$ and $y_0 < y_*$.

(d) $k = 0.0001$, $z = 0.96$, $R_0 = 6$ and $y_0 < y_*$.

**Fig 11. The comparison of the analytical solution (16) and the numerical solution of (5) around $y = 0$ as $k \to 0$.** All solutions, both analytical and numerical, are eventually converging to zero whenever $y_0 < y_*$. Hence disease elimination is possible in this case.

**Theorem 2** *Let $W_1^{*z} = (1 - y_z^*)^{-\frac{1}{k}} - 1$. The disease-free equilibrium $y = 0$ of model (18) is always locally asymptotically stable (LAS). Moreover, if $R_0 < (>)1$, $y_z^*$ is LAS (unstable), whereas $y = 1$ is unstable (LAS). A local bifurcation occurs at $y = 1$ whenever $R_0 = 1$.*

**Proof.** Let $\hat{\lambda}_z$ represent the eigenvalue of model (18). Then

$$\hat{\lambda}_z = \mu \left[ (k+1)(1-y)^{\frac{1}{k}} - k \right] [R_0 \mathcal{F}(y; k) - 1]$$

$$+ \mu k R_0 W_1 (1-y)^{\frac{k+1}{k}} \mathcal{F}'(y; k). \tag{19}$$

At $y = 0$, $\hat{\lambda}_z = -\mu < 0$ since $\mu > 0$, so $y = 0$ is LAS. For $y = 1$,

$$\hat{\lambda}_z = \mu k(1 - R_0) \begin{cases} < 0 & \text{if } R_0 > 1 \Rightarrow \quad y = 1 \text{ is LAS} \\ > 0 & \text{if } R_0 < 1 \Rightarrow \quad y = 1 \text{ is unstable} \\ = 0 & \text{if } R_0 = 1 \Rightarrow \quad y = 1 \text{ is a bifurcation point.} \end{cases}$$

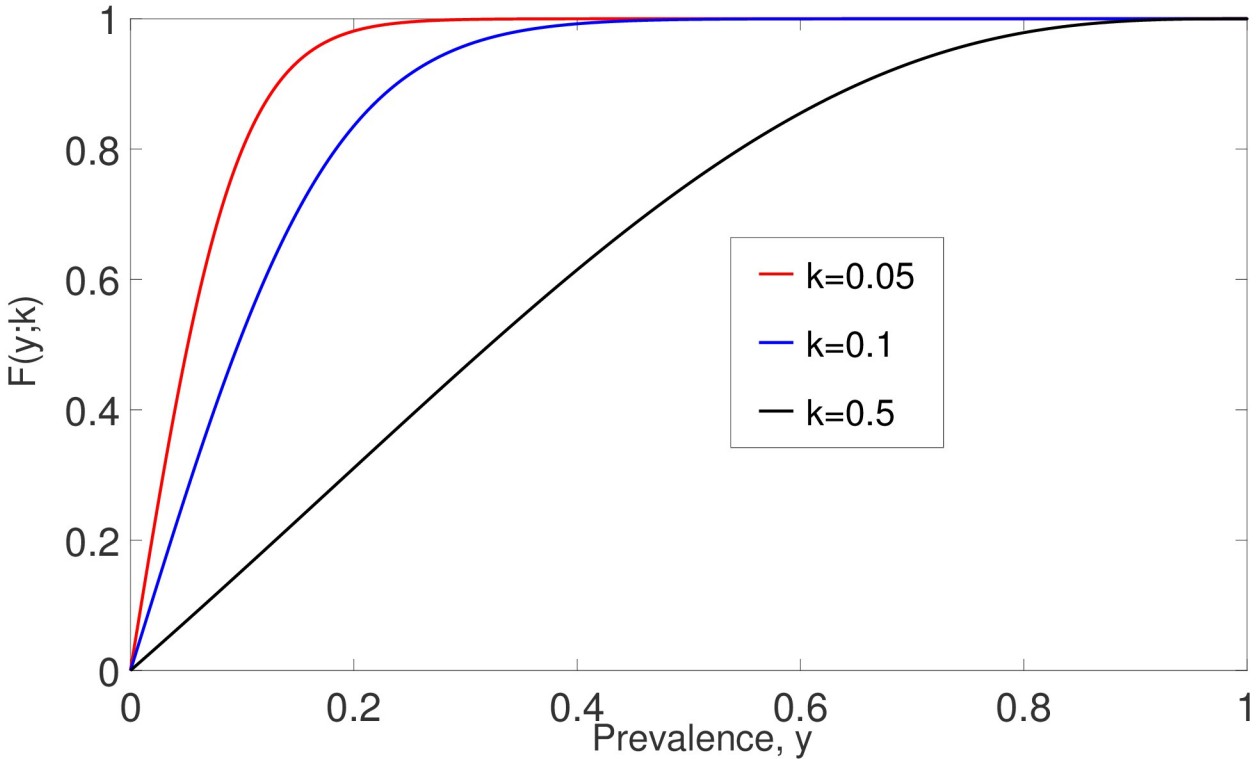

**Fig 12. The relationship between $\mathcal{F}(y; k)$ and prevalence, $y$.**

For $y = y_z^*$,

$$\hat{\lambda}_z\bigg|_{y=y_z^*} = \mu(k+1)(R_0 - 1)\left[1 - \left(\frac{R_0 - 1}{R_0}\right)^{\frac{1}{k+1}}\right].$$

Since $\mu$, $R_0$, $k$ and

$$1 - \left(\frac{R_0 - 1}{R_0}\right)^{\frac{1}{k+1}} > 0,$$

then the sign of $\hat{\lambda}_z\big|_{y=y_z^*}$ is determined by $R_0 - 1$. Thus

$$\hat{\lambda}_z\bigg|_{y=y_z^*} \begin{cases} > 0 & \text{if} \quad R_0 > 1 \Rightarrow \quad y_z^* \text{ is unstable} \\ < 0 & \text{if} \quad R_0 < 1 \Rightarrow \quad y_z^* \text{ is LAS.} \end{cases}$$

To validate Theorem 2, the dynamics of the model (18) are depicted in Fig 13 with arbitrary $k$, $R_0$ and initial values. In Fig 13(a) and 13(c), all trajectories of model (18) are converging to either zero or $y_z^*$ as $t \to \infty$ for arbitrary $R_0 < 1$. That is, both $y = 0$ and $y = y_z^*$ achieve local asymptotic stability whenever $R_0 < 1$. However, for $R_0 > 1$, all solutions of model (18) are approaching either $y = 0$ or $y = 1$ as $t \to \infty$ and a separatrix in between the $\omega$-limit sets of $y = 0$ and $y = 1$ exists since $y_z^*$ is an unstable equilibrium point in this case. Moreover, these results show that when there is no density-dependence effect acting on the parasite population—that

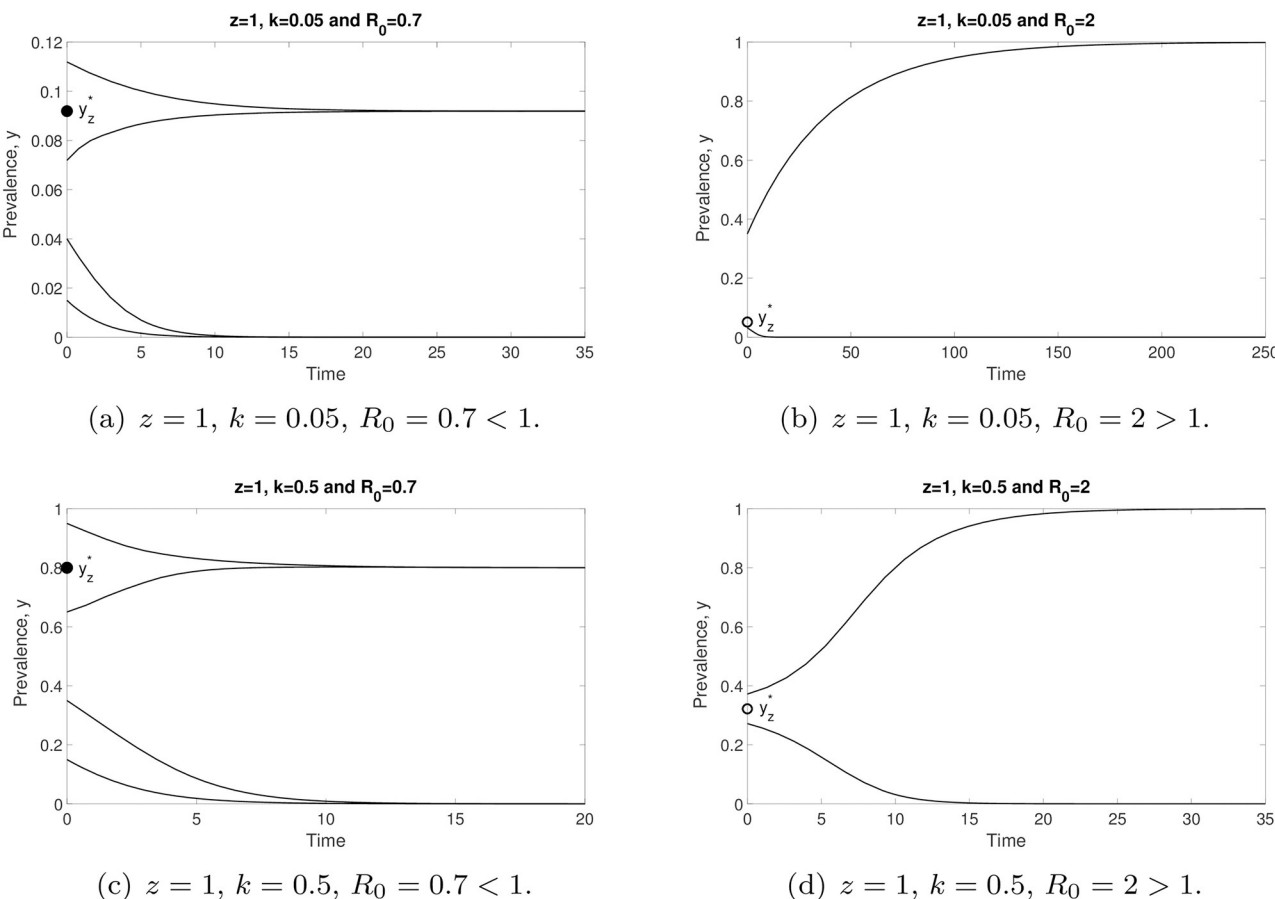

**Fig 13. The dynamics of the model (18) with arbitrary $k$, $R_0$ and initial values.** Parasite extinction is possible if the initial value of $y$ is sufficiently low. Otherwise, the disease will remain endemic.

is, even with the existence of one worm, there is a possibility for the worm to generate plenty of eggs—and if $R_0$ and the prevalence of infection at the initial stage are sufficiently high, then STH infection persists and the entire population will theoretically get infected. If the initial prevalence of infection is sufficiently low, there is a possibility that transmission will die out. In general, this illustrates the importance of density-dependent effects in the regulation of both parasitised and free-living infective worms.

Next, we approximate the solution of (18) around $y = 0$. By considering the limit $y \to 0$, we have $(1 - y)^{-\frac{1}{k}} \approx 1 + y/k$, $(1 - y)^{\frac{k+1}{k}} \approx 1 - y[(k + 1)/k]$ and

$$\mathcal{F}(y; k) \approx \frac{k + 1}{2k}\left[1 - \left(\frac{k + 2}{4k}\right)y\right]y.$$

Up to $\mathcal{O}(y^2)$ terms, the expansion of model (18) around $y = 0$ is

$$\frac{dy}{dt} = \mu y\left[\frac{(k + 1)(R_0 + 2)}{2k}y - 1\right]. \tag{20}$$

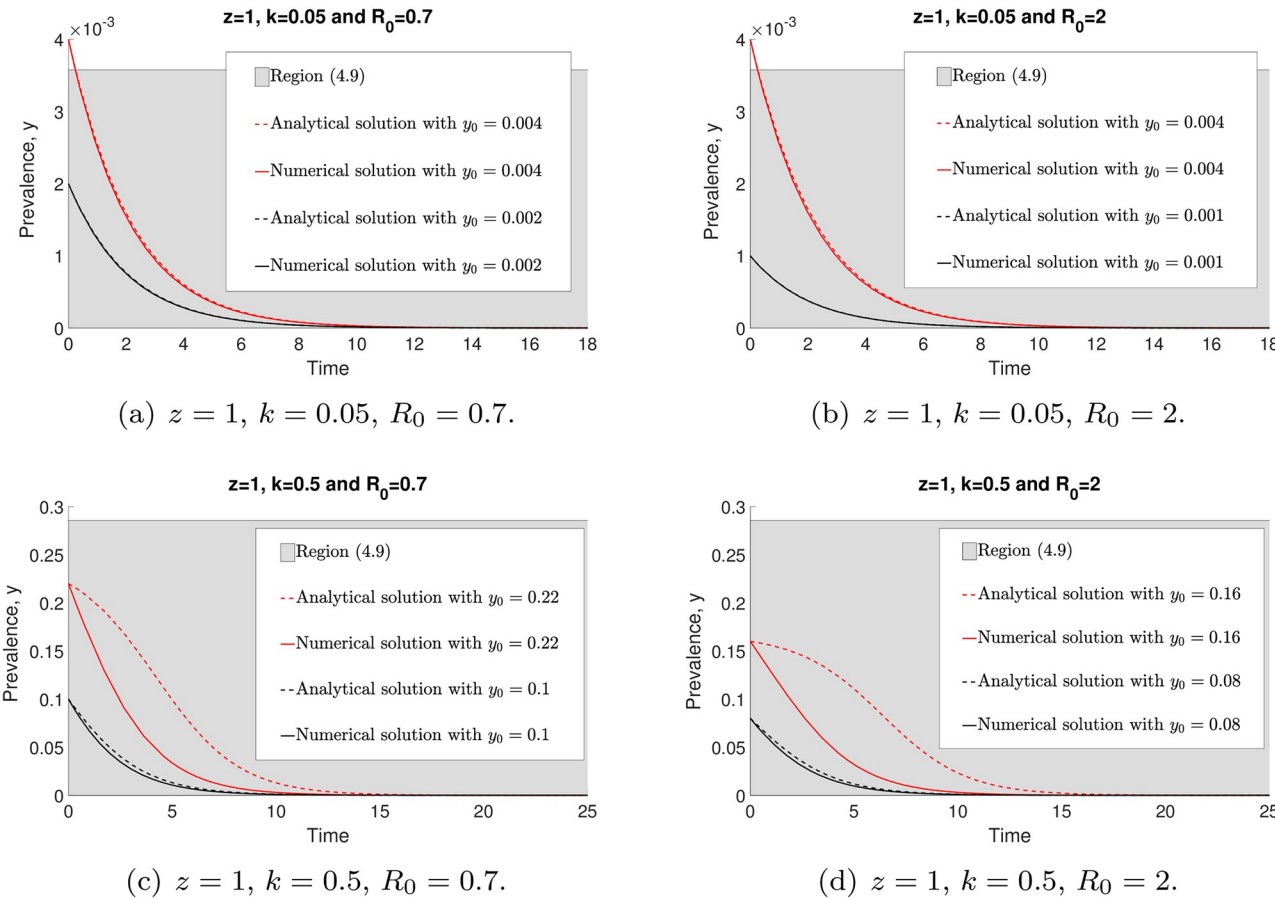

**Fig 14. The comparisons of analytical (21) and numerical (18) solutions with arbitrary $k$, $R_0$ and initial values around $y$ = 0.** Both analytical and numerical solutions are eventually converging to zero.

The analytical solution of (20) is given by

$$y(t) = \frac{2ky_0}{(k+1)(R_0+2)(1-e^{\mu t})y_0 + 2ke^{\mu t}} \,, \tag{21}$$

where $y_0$ is the initial value of $y$ and the expansion of (20) is most accurate if

$$y < \min\left\{ \frac{6k^2}{3k+2} \left| \frac{1}{k-2} \right|, \frac{3k}{2k+1}, \left| \frac{3k}{k-1} \right| \right\} \equiv y_{agr2} \,, \tag{22}$$

where $k \neq \{1, 2\}$.

The analytical (21) and numerical solutions of model (20) are depicted in Fig 14. By increasing $k$, region (22) (in grey) gets larger. From this figure, it is demonstrated that both analytical and numerical solutions are in good agreement, especially when the initial value $y_0$ is sufficiently close to zero and $k$ is small. Nevertheless, for $y_0$ in the neighbourhood of zero, these two solutions eventually approach zero for arbitrary $k$ and $R_0$ values. In conclusion, the disease-free equilibrium is locally asymptotically stable and the disease will die off whenever $y_0$ is sufficiently close to zero.

## Stochastic prevalence model

To investigate further the dynamical behaviour around the disease-free equilibrium ($y = 0$), we introduce stochastic perturbations into the deterministic model (5) and investigate how the stochastic noise affects the dynamics of the model (5) around the disease-free equilibrium (DFE). By adding demographic noise into model (5), a stochastic version of the prevalence model (5) is given as follows:

$$
\begin{aligned}
dy(t) \quad &= \mu k[1 - y(t)]\left\{1 - [1 - y(t)]^{\frac{1}{k}}\right\}[R_0 \mathcal{F}(y(t); k, z) - 1]\mathrm{d}t \\
&\quad + \rho y(t) dB(t),
\end{aligned}
\tag{23}
$$

where $\rho$ is the intensity of the Gaussian white noise and $B(t)$ is Brownian motion.

The introduction of the noise term in (23) is motivated by the sample variance induced through finite population effects. Such a term is derivable in the mean-field expansion from a master equation approach, which we do not demonstrate here. Due to the population heterogeneity introduced by aggregation in STH (and other helminth) transmission models, this term is will be accompanied by higher-order additional noise terms in the full expansion, whose derivation we leave to future work.

The sufficient condition for parasite extinction is defined in the following theorem.

**Theorem 3** *Let $\mathcal{F}_{\max} = \mathcal{F}(y_{\mathrm{bp}}; k, z)$. If $R_0 \leq 1/\mathcal{F}_{\max}$ and $\rho^2 > 2\mu k$, then the solution of* (23) *satisfies*

$$
\lim_{t \to \infty} \sup \leq \mu k - \frac{\rho^2}{2} < 0 \quad \text{almost surely.}
$$

*That is, model* (23) *predicts that the parasite will die out with probability one.*

**Proof**. By Itô's formula, we get

$$
\begin{aligned}
d(\ln y) \quad &= \left\{\frac{2\mu k(1 - y)\left[1 - (1 - y)^{\frac{1}{k}}\right][R_0\mathcal{F}(y; k, z) - 1] - \rho^2 y}{2y}\right\}dt \\
&\quad + \rho dB(t) \\
&\leq \left\{-\mu k[R_0\mathcal{F}(y; k, z) - 1] - \frac{\rho^2}{2}\right\}dt + \rho dB(t), \\
&\leq \left(\mu k - \frac{\rho^2}{2}\right)dt + \rho dB(t),
\end{aligned}
\tag{24}
$$

where $(1 - y)\left[1 - (1 - y)^{\frac{1}{k}}\right] \geq -y$ and $R_0\mathcal{F}(y; k, z) - 1 \leq 0$ if $R_0 \leq \frac{1}{\mathcal{F}_{\max}}$.

Integrating (24) from 0 to $t$ yields

$$
\ln y(t) \leq \ln y(0) + \left(\mu k - \frac{\rho^2}{2}\right)t + G_1(t), \quad \text{where } G_1(t) = \int_0^t \rho dB(\zeta).
$$

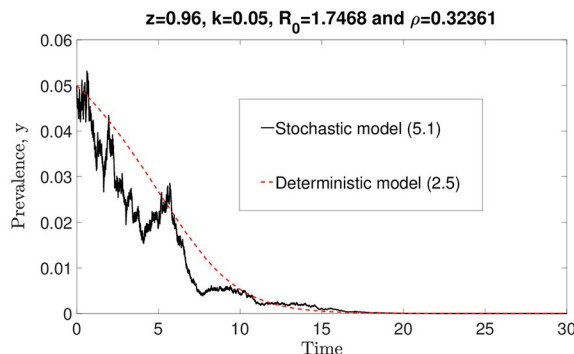

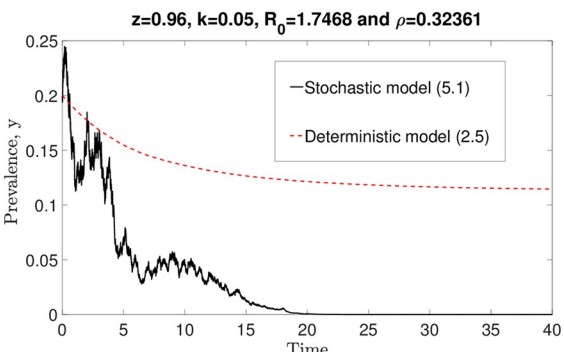

(a) $k = 0.05$, $R_0 = 1.7468$, $\rho = 0.32361$ and $y_0$ around DFE.

(b) $k = 0.05$, $R_0 = 1.7468$, $\rho = 0.32361$ and large $y_0$.

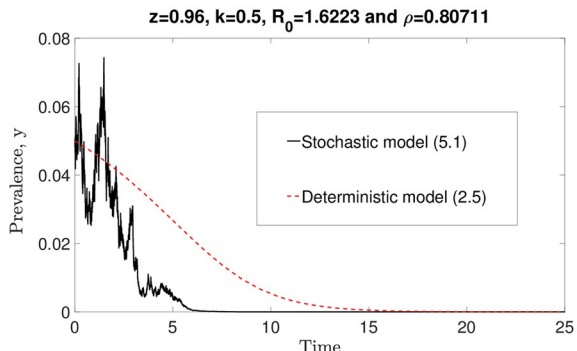

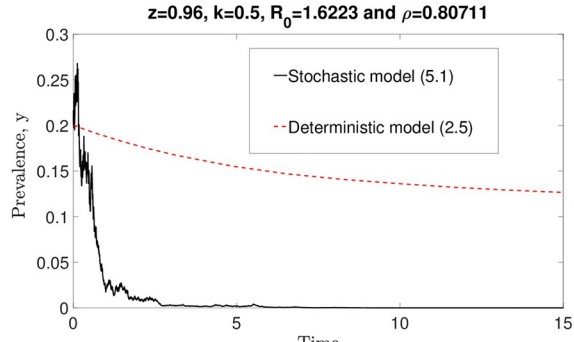

(c) $k = 0.5$, $R_0 = 1.6223$, $\rho = 0.80711$ and $y_0$ around DFE.

(d) $k = 0.5$, $R_0 = 1.6223$, $\rho = 0.80711$ and large $y_0$.

**Fig 15. Comparisons of the stochastic (23) and deterministic (5) models with arbitrary $k$, $\rho$ and $y_0$ values.** By choosing parameter values that satisfy the conditions in Theorem 3, all solutions of the stochastic model (23) with arbitrary $k$ and $y_0$ eventually converge to zero. However, if $y_0$ is sufficiently large, the stochastic and deterministic models produce conflicting results. That is, solutions of the stochastic model approach zero, whereas solutions of the deterministic model remain endemic.

Thus

$$\limsup_{t\to\infty} \frac{\ln y(t)}{t} \leq \left( \mu k - \frac{\rho^2}{2} \right) < 0 \quad a.s. \quad \text{if } \rho^2 > 2\mu k.$$

Moreover,

$$\limsup_{t\to\infty} \frac{\langle G_1, G_1 \rangle}{t} = \limsup_{t\to\infty} \frac{1}{t} \int_0^t \rho^2 d\zeta = \rho^2 < \infty$$

and, by the strong law of large numbers of martingales, $\limsup_{t\to\infty} G_1(t)/t = 0$ almost surely [38].

By selecting $z = 0.96$, $\mu = 0.5$ and parameter values that fulfil the requirements of Theorem 3, the comparisons of numerical simulation of the stochastic (23) and deterministic (5) models are depicted in Fig 15. For sufficiently small $y_0$ values, we find that both numerical solutions of the stochastic and deterministic models eventually converge to zero (see Fig 15(a) and 15(c)) and lead to the same conclusion: the infection will die out. However, both models produce contradictory results whenever $y_0$ is sufficiently large (see Fig 15(b) and 15(d)). The stochastic model (23) predicts disease eradication, but the deterministic model (5) forecasts the

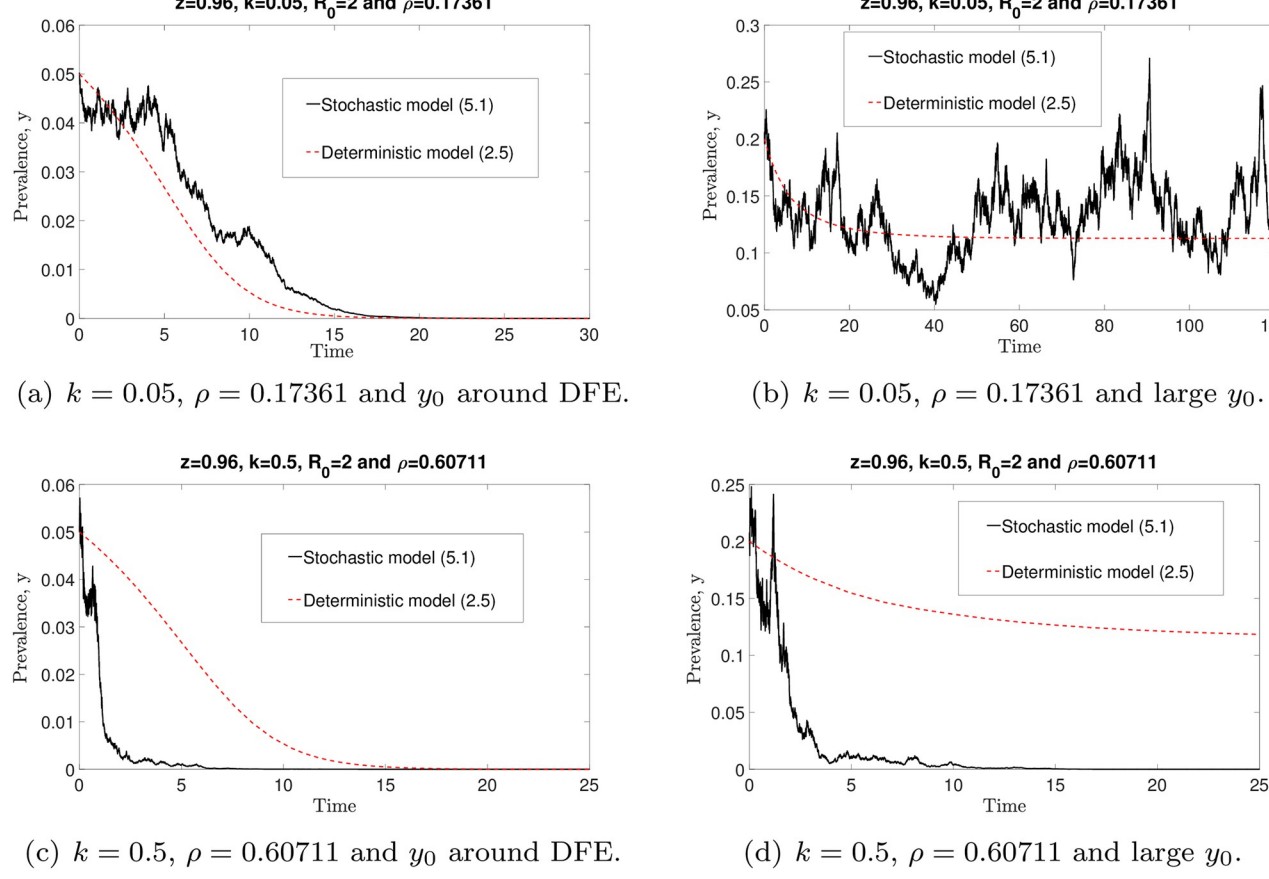

**Fig 16. Comparisons of the stochastic (23) and deterministic (5) models, with varying $k$, $\rho$ and $y_0$ values.** By choosing parameter values such that the sufficient conditions as in Theorem 3 are violated, not all solutions of the stochastic model (23) approach zero. For small $k$ and sufficiently large $y_0$ values, the solution of the stochastic model (23) fluctuates around the endemic equilibrium state (see (b)).

persistence of the infection. Nonetheless, we can see that, if all the sufficient conditions under which the infection will go extinct are satisfied, all solutions of the stochastic model (23) eventually converge to zero for arbitrary $k$, $\rho$ and $y_0$ values. These numerical results validate Theorem 3.

Conversely, by choosing $z = 0.96$, $\mu = 0.5$, $R_0 = 2 > 1/\mathcal{F}_{\max}$ and $\rho^2 < 2\mu k$ (which violate the conditions of Theorem 3), we observe that, in Fig 16(b) in particular, the solution of the stochastic model is fluctuating around the endemic equilibrium for small $k$ and sufficiently large $y_0$ values. Thus disease extinction is not guaranteed if the conditions of Theorem 3 are violated. Hence model (23) predicts that disease eradication will occur if $R_0$ and $k$ values are bounded by $1/\mathcal{F}_{\max}$ and $\rho^2/(2\mu)$, respectively.

## Discussion

Adapting the human helminth parasite transmission model with an infectious reservoir from Anderson & May [23], we have developed a novel deterministic model to investigate the transmission dynamics of STH infections in a human population, which focuses on the prevalence of infection as an easily measurable epidemiological statistic. Where there is continuous exposure to infection in the human population, no host demography changes and no intervention has taken place, we described the existence of equilibria and their stability. Analytical results

are obtained by approximation to examine movement/dynamical trajectories around the various equilibrium states. The predictions and behaviour of the simple prevalence-based model match well with the more complex macroparasite-distribution models for arbitrary $k$, $R_0$ and initial values of prevalence. As such, the simplifications embedded in the model permit greater analytical exploration.

Our theoretical work has potential applications in the real world, given current efforts to control or eradicate STH infections in regions of endemic infection by mass drug administration and improvements in clean water supply and sanitation. The TUMIKIA project in Kenya, for example, used several schemes for mass drug administration [31], but these could be tailored to different regions, depending, for example, on the intensity of worm clumping.

The second derivatives around the equilibrium states (both stable and unstable) inform how situations in the real world might behave as control measures intensify. The movement of $y$ and its implication are summarized in Table 2. In addition, we also discussed two special cases: the limit of highly aggregated parasite distribution within human communities ($k \rightarrow 0$), and the absence of density-dependent effects acting on the parasite fecundity ($z = 1$). The former informs types of behaviour that might be observed when control actions restrict infection to a few individuals who have been non-compliers to treatment. Targeting these individuals could lead to transmission interruption, but identifying them may pose many challenges.

When aggregation of STH parasites in the human host population is low (i.e., when $k$ is large), the prevalence value at the endemic state $y^*$ increases. See Fig 1, as predicted by the negative binomial model of parasite distributions within host communities [23]. Such patterns are recorded in large-scale epidemiological studies.

The eradication of parasite transmission in a defined human community, in the absence of migration in and out of the community, is possible if the aggregation of STH parasites in the host population is high. In other words, if only a few people in the population carry the vast majority of STH infection, as is sometimes observed, it is possible to eradicate the infection with highly targeted treatment of these few individuals. The challenge is of course to identify these infected individuals, which may be costly in terms of implementing well-structured monitoring and evaluation programmes.

The eradication of transmission is not always possible, particularly if we assume that there are no, or limited, density-dependent effects acting on parasite fecundity (i.e., when $\gamma = 0$, it follows that $z = 1$). For this case, STH infection will stay in the endemic state if the initial value of prevalence is sufficiently large. Otherwise, the infection may die out. The clumping parameter $k$ and the initial value of prevalence have a significant influence on the likelihood of parasite eradication.

**Table 2. The movement of $y$ and its implications.**

| Prevalence | $0 < y < y_*$ | $y_* < y < y^*$ | $y^* < y < 1$ |
|---|---|---|---|
| Velocity | Negative | Positive | Negative |
| Acceleration | Increasing from negative values (around $y = 0$) to zero and then to positive values (around $y_*$). | Decreasing from positive values (around $y_*$) to zero and then to negative values (around $y^*$). | Increasing from negative values (around $y^*$) to zero and then to positive values (around $y = 1$). |
| Movement of solution $y$ | For $y < y_*$, $y$ moves towards $y = 0$ slowly, but its movement is speeding up when it moves sufficiently close to $y = 0$. | Solution $y$ moves away from $y_*$ quicker and eventually approaches $y^*$ with slower speed. | Solution $y$ moves towards $y^*$ slowly from the neighbourhood of $y = 1$, but its movement is speeding up when it is approaching $y^*$. |
| Implication | The elimination of STH is possible if the prevalence value can be suppressed below $y_*$. | STH infection remains endemic whenever $y > y_*$. Hence control strategies or treatment are required in order to lower the prevalence or eradicate the disease. | Similar implications as in the case of $y_* < y < y^*$. |

To investigate the impact of stochastic perturbation in the transmission dynamics of STH infection, especially around the disease-free equilibrium, stochastic noise was added into the deterministic model (5). Sufficient conditions for the extinction of the infection were identified and the numerical solutions of the stochastic and deterministic models compared. The stochastic model (23) predicts that disease extinction is certainly possible (even if $y_0$ is large) if $R_0$ and $k$ values are bounded by $1/\mathcal{F}_{\max}$ and $\rho^2/(2\mu)$, respectively.

The models analysed here have several limitations, which should be acknowledged. We have made the following key assumptions: namely, there is continuous exposure to infection by STHs (with a force of infection as in the Anderson and May model [23], which is constant with host age), that no intervention has previously been applied, that the total human population remains constant and that the dynamics of the infectious reservoir of eggs or larvae operate on a sufficiently fast timescale so that the reservoir is in quasi steady state.

This work is a preliminary study of the properties of prevalence-based macro-parasite models to help explore the transmission dynamics of STH infection in a human population by considering continuous infection using prevalence of infection as the most easily measurable epidemiological quantity. Most WHO guidelines for STH control and monitoring and evaluation programmes employ prevalence of infection as the key outcome variable. The intensity of infection is much more difficult to measure reliably in field-based studies. In future work, the prediction abilities of the prevalence-based model described in this paper could be improved by considering age structure in the human population. This would permit examining the application of interventions and assessing the effectiveness of different control strategies where treatment with drugs or behavioural change varies between different age groupings such as pre-school-aged children, school-aged children and adults. These three age groupings are commonly used to define who should be treated in MDA programmes. As control efforts intensify, human movement patterns will become important, given spatial heterogeneity in infection levels and drug coverage, as observed by health-intervention units. As such, spatially structured models are also an important expansion in future research for the prevalence-based model structures outlined in this paper.

## Supporting information

**S1 Appendix. Bifurcation point of model (5).**
(PDF)

## Acknowledgments

For citation purposes, please note that the question mark in "Smith?" is part of her name.

## Author Contributions

**Conceptualization:** Roy M. Anderson.

**Formal analysis:** Nyuk Sian Chong.

**Funding acquisition:** Roy M. Anderson.

**Investigation:** Nyuk Sian Chong, James E. Truscott.

**Methodology:** Nyuk Sian Chong, Stacey R. Smith?, James E. Truscott.

**Software:** Robert J. Hardwick.

**Supervision:** Roy M. Anderson.

**Writing – original draft:** Nyuk Sian Chong.

**Writing – review & editing:** Stacey R. Smith?.

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
