## [Decision Letter · Decision Letter 0]

4 Jul 2022

PONE-D-21-39944A prevalence-based transmission model for the study of the epidemiology and control of soil-transmitted helminthiasisPLOS ONE

Dear Dr. Smith?,

Thank you for submitting your manuscript to PLOS ONE. After careful consideration, we feel that it has merit for publication after a minor revision. Therefore, we invite you to submit a revised version of the manuscript that addresses the points raised during the review process.

Note that it is a minor revision so please consider making the suggested changes.

We look forward to receiving your revised manuscript.

Kind regards,

Pablo Martin Rodriguez

Academic Editor

PLOS ONE

Journal Requirements:

Reviewers' comments:

Reviewer's Responses to Questions

**Comments to the Author**

1. Is the manuscript technically sound, and do the data support the conclusions?

Reviewer #1: Yes

Reviewer #2: Yes

2. Has the statistical analysis been performed appropriately and rigorously? 

Reviewer #1: N/A

Reviewer #2: Yes

3. Have the authors made all data underlying the findings in their manuscript fully available?

Reviewer #1: Yes

Reviewer #2: Yes

4. Is the manuscript presented in an intelligible fashion and written in standard English?

Reviewer #1: Yes

Reviewer #2: Yes

5. Review Comments to the Author

Reviewer #1: The paper is valuable and deserves to be published after some minor textual improvements.

ABSTRACT/TEXT

(1) The purpose of the study is not in the abstract. Usually, standard formulations of STH transmission are not defined in terms of prevalence, but some of the authors have published papers on that. In fact, a previous work [23] on this subject is the starting point of the paper. This way, it is not clear in the abstract what are the novelties of the present work.

(2) The same previous remark is also valid for some parts of the main text. The authors claim (lines 78-80) that their objective is to provide some analytical insights into the transmission dynamics of STH. This in fact is achieved throughout the paper, but the general idea of these insights is not provided in the introduction. What are the new and unique aspects of the work?

(3) Could the authors be more precise and clear in the last sentence of the abstract? What do you mean by "complicated"? Please try to refer to the word "distribution" and explain it in more general terms for those who do not come from the field of applied mathematics.

"It follows that control of soil-transmitted helminths will be **complicated** by the heterogeneity of worm **distribution**."

MAIN TEXT

(4) The authors argue (lines 288-290) that their theoretical work has many applications. However, the precise meaning of applications is lacking here.

Minor issue:

(5) After Equation 3, please provide a reference for the described R0 (reference [23], for example).

Reviewer #2: The paper on prevalence-based transmission model for the study of the epidemiology and control of soil-transmitted helminthiasis, provides realistic insights into the soil-transmitted helminthiasis. The paper is well motivated, and its objectives are achieved via the use of mathematical modelling and data analytic approaches. The developed in the paper, which typically take the form of deterministic system of nonlinear differential equations, are interesting. Further, some of the theoretical and data analytics approaches adopted are equally interesting. I therefore recommend the paper for publication.

6. PLOS authors have the option to publish the peer review history of their article (what does this mean?). If published, this will include your full peer review and any attached files.

Reviewer #1: **Yes: **Diego Samuel Rodrigues

Reviewer #2: **Yes: **Evans Otieno Omondi

---

## [Editor Report · Decision Letter 1]

25 Jul 2022

A prevalence-based transmission model for the study of the epidemiology and control of soil-transmitted helminthiasis

PONE-D-21-39944R1

Dear Dr. Smith?,

We’re pleased to inform you that your manuscript has been judged scientifically suitable for publication and will be formally accepted for publication once it meets all outstanding technical requirements.

Kind regards,

Pablo Martin Rodriguez

Academic Editor

PLOS ONE

---

## [Editor Report · Acceptance letter]

3 Aug 2022

PONE-D-21-39944R1 

A prevalence-based transmission model for the study of the epidemiology and control of soil-transmitted helminthiasis 

Dear Dr. Smith?:

I'm pleased to inform you that your manuscript has been deemed suitable for publication in PLOS ONE. Congratulations! Your manuscript is now with our production department. 

Kind regards, 

on behalf of

Professor Pablo Martin Rodriguez 

Academic Editor

PLOS ONE